# Potent AMA1-specific human monoclonal antibody against *Plasmodium vivax* Pre-erythrocytic and Blood Stages

Anna C. Winnicki[1,16], Melanie H. Dietrich [2,3,16], Lee M. Yeoh [4,5], Lenore L. Carias[1], Wanlapa Roobsoong [6], Chiara L. Drago[4,7], Alyssa N. Malachin[1], Karli R. Redinger [1], Lionel Brice Feufack-Donfack[8], Lea Baldor[8], Nicolai C. Jung[2], Olivia S. McLaine[1], Yelenna Skomorovska-Prokvolit[1], Agnes Orban[8], D. Herbert Opi [4,9,10], Payton Kirtley[11], Kiersey Nielson [11], Maya Aleshnick[11], Gigliola Zanghi[12], Nastaran Rezakhani[12], Ashley M. Vaughan [12,13], Brandon K. Wilder [11], Jetsumon Sattabongkot[6], Wai-Hong Tham [2,3], Jean Popovici [8], James G. Beeson[4,9,10], Jürgen Bosch [1,14,17] ✉ & Christopher L. King [1,15,17] ✉

New therapeutics are necessary for preventing *Plasmodium vivax* malaria due to easy transmissibility and dormancy in the liver that increases the clinical burden due to recurrent relapse. In this manuscript we characterize 12 Pv Apical Membrane Antigen 1 (PvAMA1) specific human monoclonal antibodies from Peripheral Blood Mononuclear Cells of a Pv-exposed individual. PvAMA1 is essential for sporozoite and merozoite invasion, making it a unique therapeutic target. We show that humAb 826827 blocks the invasion of human reticulocytes using Pv clinical isolates and inhibits sporozoite invasion of human hepatocytes in vitro (IC$_{50}$ of 0.3 – 3.7 μg/mL). Inoculation of human liver transgenic (FRG-humHep) female mice with humAb 826827 significantly reduces liver infection in vivo. The crystal structure of rPvAMA1 bound to 826827 shows that 826827 partially occupies the highly conserved hydrophobic groove in PvAMA1 that binds its known receptor, RON2. We have isolated a potent humAb that is isolate-transcendent, blocks both pre-erythrocytic and blood stage infection, and could be a potential therapy for Pv.

Half of the world's population is at risk of malaria, with 247 million cases and 619,000 deaths occurring in 2022[1]. *Plasmodium falciparum* (Pf) and *Plasmodium vivax* (Pv) account for most human malaria cases. Pf predominates in sub-Saharan Africa, whereas Pv accounts for 80% of malaria in Asia and the Americas[2]. Pv causes significant disease, especially in low-middle-income countries burdened with poor nutrition, anemia, and co-infections[3]. During the erythrocytic or blood stage, Pv merozoites infect reticulocytes, primarily at the sites of erythropoiesis in the bone marrow and spleen[4-6]. Pv also forms a dormant pre-erythrocytic or liver phase (hypnozoites), causing frequent relapses,

further contributing to anemia and other complications[7,8]. The frequent relapses produce gametocytes that drive transmission in populations[9]. Presently, there is no vaccine available for Pv. Therefore, targeted therapies that inhibit hepatocyte and erythrocyte invasion are crucial in reducing the overall disease burden and enabling Pv elimination[10-13].

An essential piece of the invasion machinery used by sporozoites and merozoites is the interaction between Apical Membrane Antigen 1 (AMA1) and an extracellular ß-hairpin loop in the C-terminal portion of Rhoptry Neck Protein 2 (RON2)[14]. Utilized by all members of the

Apicomplexa, AMA1 originates in the parasite's micronemes and is later translocated to the parasite's membrane. In human hosts, AMA1 is expressed during the late merozoite and sporozoite stages[15–17]. Immediately preceding invasion, the RON complex (composed of RON2, 4, 5, and 8), is secreted from the parasite's rhoptries and embeds into the target cell's membrane. The extracellular ß-hairpin loop near the C-terminus of RON2 can then interact with AMA1 by nestling into the hydrophobic groove of Domain 1 of AMA1[18]. For the RON2-loop to properly engage with AMA1, a mobile loop of Domain 2 of AMA1 is displaced by the incoming ligand to reveal the entire hydrophobic groove[16,19]. This protein-protein interaction is important for forming a tight junction that allows the merozoite to move across the extracellular space and into the erythrocyte through connections between the parasite surface and its myosin motor along actin filaments[7,14,15,20–22]. Blocking the interaction of PfAMA1 or PkAMA1 with PfRON2 or PkRON2 by antibodies[14,23–25] or peptides[14,26,27] inhibits invasion, confirming that the interaction of AMA1-RON2 is important for the *Plasmodium* life cycle. While blocking this interaction can prevent invasion, there is evidence that AMA1 interacts directly with the surface of the erythrocyte[28] or with other receptors[29], which are yet to be defined.

Malaria-infected individuals acquire partial immunity to infection and disease primarily directed toward blood-stage parasites[30]. Antibodies play a key role in this partial immunity, as demonstrated by transferring human IgG from immune adults to non-immune children, resulting in protection from malaria. The relative contribution of different malaria blood-stage antigens to producing Pv- and Pf-specific antibodies is yet to be fully understood. The antibodies may protect against malaria by one or more mechanisms: 1) blocking merozoite invasion into erythrocytes, 2) complement activation, and 3) opsonic phagocytosis by monocytes and neutrophils[31,32]. Elevated antibodies to specific *Plasmodium* antigens, such as AMA1, are associated with protection against infection and disease[33–35].

PvAMA1, a three-domain protein, is under immune selection, posing a problem for developing vaccines and therapeutic monoclonal antibodies to this antigen[32,36–38]. Domains 1 and 2 contain two clusters of disulfide-bonded cysteines, with the crystal structure revealing regions comprised of many long loops. Extending from the core of Domain 1, these long loops allow for significant variation and protein flexibility. This aids parasite evasion of AMA1-specific protective human antibody responses[39]. The loops form a scaffold for the numerous polymorphisms on the surface of AMA1[34,40–44]. The hydrophobic RON2 binding groove is highly conserved across Pv clinical isolates (97.8% over 110 residues). However, this groove is surrounded by highly polymorphic residues, presumably due to selective pressure from host immune responses[45]. Since the AMA1:RON2 interaction plays a role at multiple points of the parasite's life cycle, including sporozoite infection of the liver and merozoite invasion of erythrocytes, the latter of which is essential for gametogenesis and thus transmission[8,15–17], it presents an opportunity for a multi-stage target.

Recently, the development of therapeutic monoclonal antibodies to sporozoite antigens protects against malaria in endemic populations[46]. Human monoclonals (humAbs) to Pf circumsporozoite protein (CSP) have been isolated from individuals following immunization with attenuated sporozoites[47,48]. The administration of humAbs reduced the risk of Pf infection by 88% in adults and protected against illness by up to 77% in children residing in malaria-endemic areas of Africa[19,49]. These findings suggest that utilizing human monoclonal antibodies specific to sporozoite antigens could be a promising approach to preventing malaria infection and associated illnesses. Additionally, a human-derived monoclonal antibody targeting PfAMA1 has been identified, characterized, and shown to exhibit blocking activity at an $IC_{50}$ of 35 μg/mL[50] for blood-stage parasites in vitro. To our knowledge, no human-derived monoclonal antibody specific to PvAMA1 has been documented.

We have found 12 human-derived monoclonal antibodies (humAbs) that target PvAMA1. These antibodies were taken from the blood cells of a person from Cambodia who had a history of Pv infection. They have been shown to stop PvAMA1 from binding to PvRON2. We have thoroughly studied the physical properties of these 12 humAbs. We tested their ability to stop the growth and invasion of merozoites and sporozoites in vitro. Additionally, we looked at how well our most effective humAb, referred to as 826827, could stop Pv infection in a chimeric FRG-humHep mouse model. In short, we have found a powerful humAb that can stop RON2-loop binding and consistently prevent blood stage and sporozoite infection in both in vitro with multiple clinical isolates and in vivo. Our structural data shows that this humAb attaches to the RON2 hydrophobic binding groove and targets conserved amino acids in its epitope.

## Results

### Isolation and expression of human monoclonal antibodies

We tested plasma from seven malaria-exposed Cambodian adults for antibodies that could inhibit the binding of PvRON2 to PvAMA1[29]. Using the donor with the highest RON2 binding inhibition activity (Supplemental Fig. 1), we isolated 157 PvAMA1-specific B cells (Supplemental Fig. 2) that were PCR-amplified and sequenced for the immunoglobulin heavy chain (IGH) V-D-J region. B cells were placed into 67 clonal groups based on the same VDJ segment, CDR3 length, and 85% or greater amino acid similarity for CDR3 (Supplemental Table 1). We generated 12 human PvAMA1-specific monoclonal antibodies that represent 12 clonal groups. The clones selected for the generation of humAbs were based on whether there was a corresponding immunoglobulin light chain (IGL), the quality of sequence, the degree of somatic hypermutations (SHM), and the selection of one B cell IGH + IGL pair from a clonal group (Fig. 1A, B). The humAb are named with the first 3 numbers referring to IGH and the last three IGL. For example, humAb 826827 comes from the 826-heavy chain and 827-light chain.

To determine the specificity of 12 PvAMA1 humAbs, we tested the binding capabilities to recombinant AMA1 in a multiplex immunoassay format by coating magnetic microbeads with recombinant proteins corresponding to two strains of PvAMA1 (Palo Alto, the variant used to sort B cells and PvAMA1_PNG16, a sequence with significant polymorphisms compared to Palo Alto), PkAMA1, PfAMA1_3D7, and TgAMA1 (Fig. 1C). The sequence identity of the AMA1 constructs with respect to PvAMA1_Palo Alto was 97.2%, 85.6%, 60.3%, and 30.3% for PvAMA1_PNG16, PkAMA1, PfAMA1_3D7, and TgAMA1 respectively. All humAbs recognized PvAMA1_Palo Alto, ten recognized PvAMA1_PNG16, seven recognized PkAMA1, two recognized PfAMA1_3D7, and none recognized TgAMA1 in a multiplex immunoassay[51]. The isotype control humAb 043038, specific to tetanus toxoid C-terminal fragment, did not bind to any AMA1 recombinant proteins (Fig. 1C). The humAb titers recognizing the different recombinant AMA1 constructs varied among the different humAbs and were associated with differences in humAb avidity (Fig. 1C, D). We measured avidity using the chaotropic reagent $NH_4SCN$. The calculated Avidity Index 50 ($AI_{50}$) represents the molar concentration of the chaotropic reagent where 50% of the binding of the humAb to AMA1 is lost (Supplemental Table 2). Higher $AI_{50}$ represents stronger binding. The humAbs 806807, 826827, 832833, and 838839 had the highest avidity. Of note, humAb 826827 has a higher avidity for PkAMA1 than PvAMA1. To further characterize the biophysical properties of the humAbs, we measured the affinity to PvAMA1_Palo Alto using Surface Plasmon Resonance (SPR). Affinities of humAbs to PvAMA1 ranged from $10.7 × 10^{-9}$ to $47.7 × 10^{-9}$ M (Fig. 1E, Supplemental Fig. 3). We could not accurately determine the $K_D$ for three humAbs, 800801, 804805 and 808809. The PvAMA1 humAbs had various biophysical characteristics that may have further therapeutic potential.

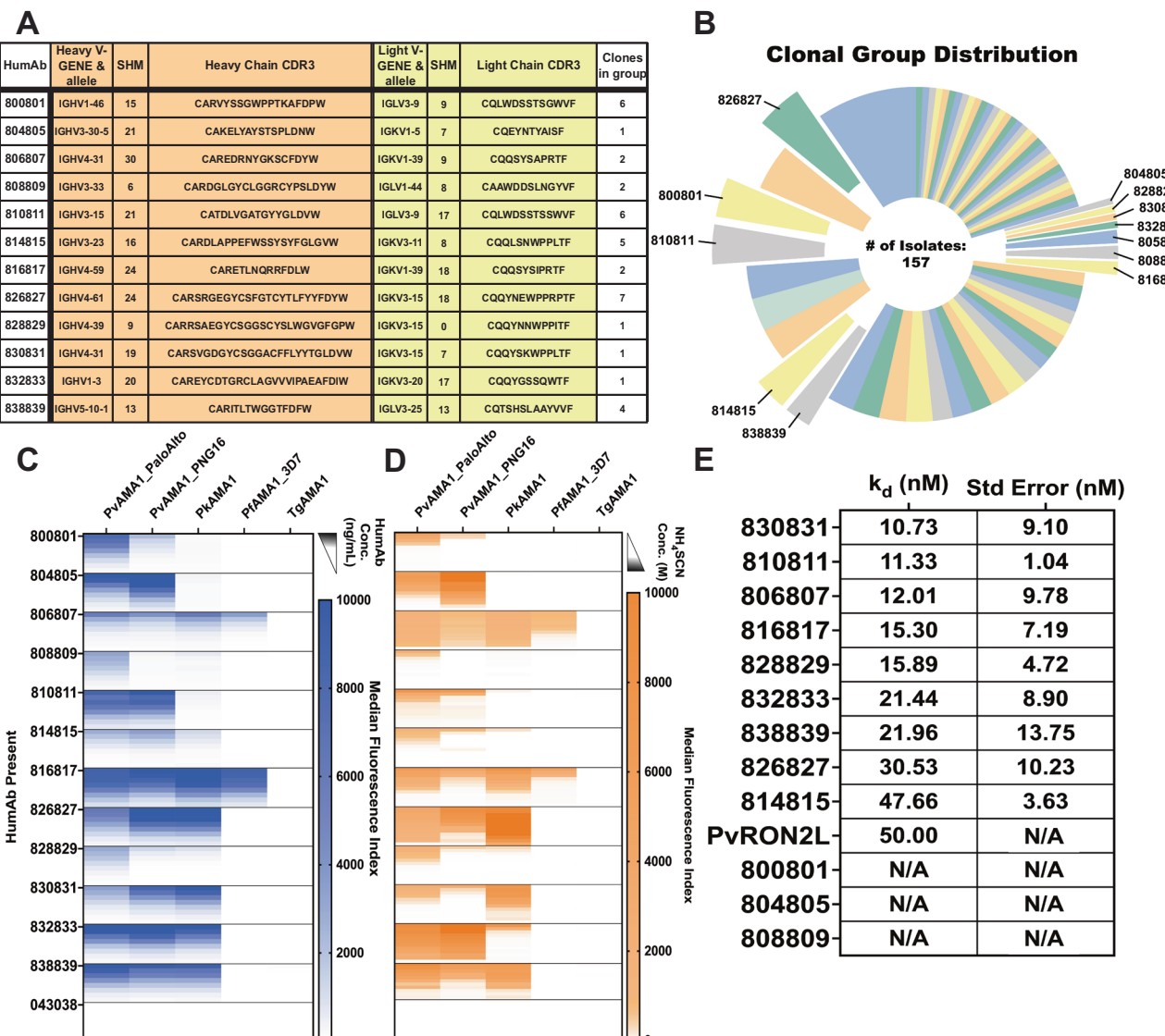

**Fig. 1 | Sequence characterization of 12 humAbs and their selectivity, avidity, and affinity towards AMA1. A** Sequences of CDR3 IGH (orange) and corresponding IGL or IGK (yellow) from individual B cells from which humAbs were generated. The number of somatic hypermutations (SHM) of nucleotides that differ from germline sequences is shown for each clone. **B** 67 clonal groups were identified, from which clonal groups PvAMA1-specific humAbs were isolated is indicated. **C** HumAb reactivity to PvAMA1_Palo Alto, PvAMA1_PNG16, PkAMA1, PfAMA1_3D7, and TgAMA1 at varying concentrations (1.0, 0.5, 0.250, 0.125, 0.062, 0.313, 0.016, 0.008 µg/mL). **D** HumAb avidity as measured by reduction in binding (MFI) to AMA1 with varying concentrations of $NH_4SCN$ (0.0, 0.9, 1.2, 1.5, 1.8, 2.1, 2.4, 2.7, 3.0, 3.3, 3.6, 4.0 M). HumAbs were used at a concentration of 0.2 µg/mL. **E** Affinities ($K_D$s) of PvAMA1-specific humAbs determined using SPR single-cycle kinetics. HumAbs 800801, 804805, and 808809 affinities could not be determined (N/A) in this assay. Standard error of the mean was calculated using Prism.

## HumAb Inhibition of Pf-PvAMA1 Transgenic Parasites

Because long-term in vitro culture of *P. vivax* is not currently possible, we used a modified Pf parasite line that can express PvAMA1[28,29,52] as a model to assess the ability of the humAbs to inhibit merozoite invasion. Figure 2A shows the $IC_{50}$ curves of the four humAbs, 808809, 826827, 828829, and 830831. These humAbs demonstrated the lowest $IC_{50}$s (2.6–11.5 µg/mL). Other humAbs targeting PvAMA1 showed higher $IC_{50}$ values or did not show blocking activity in this assay, e.g., 814815 and 816817 (Supplemental Fig. 4). HumAb 043038 was employed as a negative control and did not inhibit invasion at the tested concentrations. This assay shows humAb 826827 was the most potent, with an $IC_{50}$ of 2.6 µg/ml.

## HumAb inhibition of Pv clinical isolates from Cambodia

To assess the ability of humAbs to PvAMA1 to inhibit Pv clinical isolates, we used in vitro reticulocyte invasion assays. This assay combines schizont-enriched red cells from *P. vivax* infected subjects with enriched reticulocytes from human cord blood donors and cultured for ~10 hours to allow schizonts to rupture and release merozoites to invade reticulocytes in the presence of various humAbs. The results of this assay are shown in (Fig. 2B). Only humAb 826827 significantly inhibited Pv invasion of reticulocytes compared to the control humAb 043038 (67.4% (±8.6 SEM) vs 4.7% (±3.8 SEM), one-way ANOVA, $P = < 0.0001$). We conducted dose-response invasion assays using humAb 826827 ranging from 7.8 to 1000 µg/mL against four additional Pv clinical isolates to determine the $IC_{50}$. The average $IC_{50}$ obtained from four clinical isolates was 48 µg/mL (±6.6 SEM) (Fig. 2C). These results and the in vitro Pf-PvAMA1 transgenic parasite experiments suggest that human 826827 is highly effective in inhibiting AMA1-dependent erythrocyte invasion.

## HumAb inhibition of Pv sporozoites in human hepatocytes

We examined humAb ability to inhibit *Pv* sporozoite invasion of hepatocytes in vitro (Fig. 3). Sporozoites were isolated from the

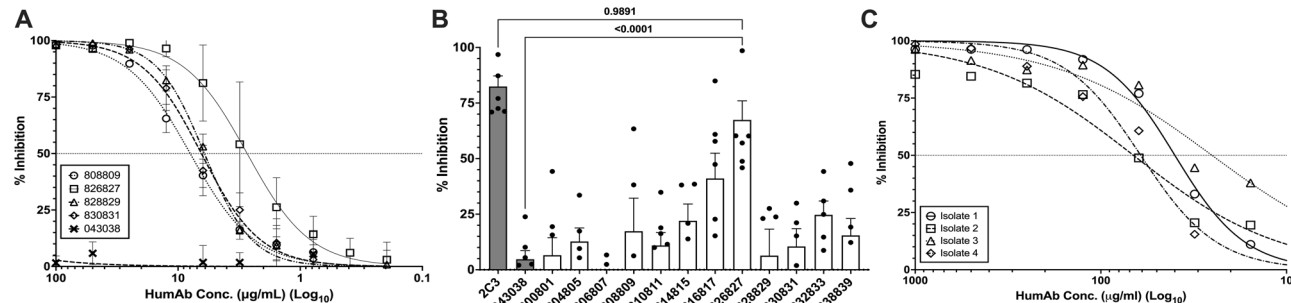

**Fig. 2 | HumAbs inhibition of blood-stage infection. A** Dose response of PvAMA1-specific humAbs against Pf-PvAMA1 transgenic parasites of humAb with lowest IC$_{50}$s. Each symbol represents the average of three replicates for every concentration. **B** The mean percentage (± SEM) of reticulocytes infected using Pv clinical isolates in short-term invasion inhibition with different humAbs at 100 μg/mL. Each dot represents a biological replicate from a different clinical isolate ($n = 2-7$). The flow cytometry background of target cells (reticulocytes) without parasites (mean is 9%, range 5–15% invasion) was subtracted from each experiment. Mouse mAb, 2C3 (100 μg/mL) binds to Duffy antigen on reticulocytes, thus blocking Pv invasion of reticulocytes (positive control) ($p$-value = 0.9891). Only humAb 826827 significantly inhibited reticulocyte invasion compared to the negative control ($p$-value = <0.0001). HumAb 043038 was used as a negative control for experiments represented in panel **A** and **B**. A multi-variant one-way ANOVA and Tukey's secondary test was used to calculate the $P$-values compared to the positive and negative controls using Prism. **C** Dose response of humAb 826827 against four different Pv clinical isolates in short-term invasion inhibition cultures (Isolate 1, 2, 3, and 4 have IC$_{50}$s of 39.96, 66.78, 25.22, and 61.04 μg/mL respectively. Note: These are different isolates than those used for Fig. 2B).

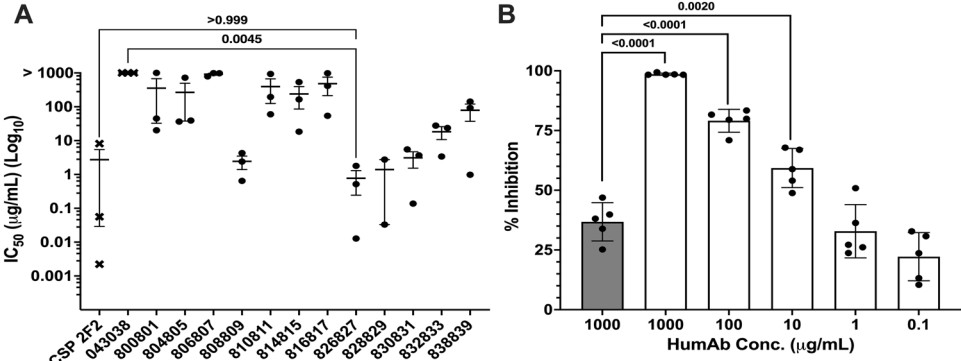

**Fig. 3 | PvAMA1-specific humAbs inhibition of sporozoite invasion of human hepatocyte HC04 cell line and primary human hepatocytes. A** IC$_{50}$ of different humAbs for Pv sporozoite invasion of human HC04 hepatocytes was performed at five concentrations (0.1–1000 μg/mL). Values represent the mean (SEM) of three biological replicates, with each biological replicate performed in duplicate. 043038 was used as a negative control ($p$-value = 0.0045). Murine anti-CSP 2F2 was used as a positive control ($p$-value = >0.999). A multi-variant one-way ANOVA and Tukey's secondary test was used to calculate the $P$-values compared to the negative control using Prism. **B** Percent inhibition of isolated Pv sporozoites ($n = 5$) into human primary hepatocytes using 826827 (white bars) at various concentrations. 043038 (shaded bar) was used as a negative control and was only tested at one concentration ($p$-value = <0.0001 at the same concentration). A multi-variant one-way ANOVA and Tukey's secondary test was used to calculate the $P$-values compared to the negative control using Prism.

salivary glands of *Anopheles dirus* mosquitoes fed on blood collected from Pv-infected subjects from Thailand, and humAbs were tested for their ability to inhibit sporozoite invasion of HC04 hepatocytes. Five humAbs showed IC$_{50}$s (0.38–2.6 μg/mL) that were comparable to the positive control anti-CSP murine monoclonal 2F2 (IC$_{50}$ CSP210 = 0.2 μg/mL, Fig. 3A, Supplemental Fig. 5[53]). For three biological replicates of humAb 826827 (Fig. 3A), IC$_{50}$s ranged from less than 0.07 to 1.8 μg/mL. We also performed the *Pv* sporozoite invasion assay on a Cambodian isolate using primary human hepatocytes and the humAb 826827 (Fig. 3B). In this experiment, the IC$_{50}$ was 3.7 μg/mL. Thus, humAb 826827 can inhibit AMA1-dependent sporozoite invasion from multiple clinical isolates.

To assess whether humAb 826827 inhibited liver stage infection in vivo, we used liver chimeric mice (FRG-humHep), which have been transplanted with human hepatocytes and support *P. vivax* sporozoites infection and liver stage development[54]. FRG-humHep mice were inoculated intravenously with 30 μg or 300 μg of humAb 826827 or 300 μg of negative control anti-tetanus toxoid humAb 043038 three hours before challenge with 400,000 freshly dissected *P. vivax* sporozoites. The average serum mAb concentration in the 300 μg 826827-treated group was 27.9 μg/mL 2 days after the sporozoite challenge and 4.1 μg/mL in the 30 μg-treated group (Fig. 4A). To determine the effect of passive immunization on liver infection, animals were sacrificed on 9 days post-infection. This represents the peak of liver-stage growth and the beginning of schizont egress from the liver. Mice that received 300 μg humAb 826827 showed a significant reduction in liver-stage parasites assessed by 18S quantitative RT qPCR compared to control humAb (Fig. 4B). However, 18S copies detected in the liver exhibited more variability in the 30 μg humAb 826827 treatment group, with one mouse showing comparable 18S copies to that observed in animals treated with 300 μg humAb 826827. The variability in 18S copies was not associated with the serum concentration of humAb 826827 in this treatment group. It is possible that some dead sporozoites remain in the liver by day 9 and their residual DNA is being detected due to the sensitivity of 18 s RT qPCR, accounting for the signal shown in the 300 μg humAb 826827 treatment group as previously seen in this model[54]. There was also a significant reduction in microscopically identified merozoite or hypnozoites in the liver (Supplemental Fig. 6). Therefore, 300 μg humAb 826827 administration significantly reduced sporozoite invasion of hepatocytes in vivo.

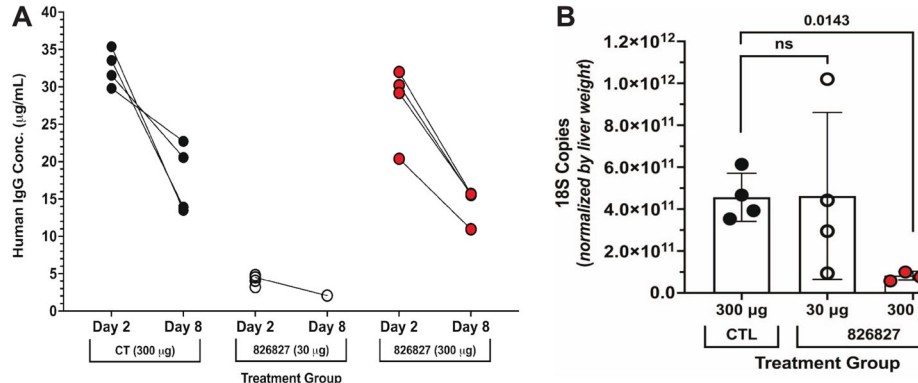

**Fig. 4 | Reduction in *P. vivax* liver infection in FRG-humHep mice after PvAMA1 monoclonal blockaid.** Mice were injected intravenously with 30 μg and 300 μg of anti-PvAMA1 (humAb 826827, *N* = 4 for each concentration) and 300 μg anti-tetanus toxoid (humAb 048038, *N* = 4) approximately 3 hours before infection with 400,000 freshly dissected *P. vivax* sporozoites. **A** Serum concentrations of humAbs were measured 2 days after the sporozoite challenge and on day 8. **B** Liver sections were harvested on day 9 post-infection, weighed, homogenized, and *P. vivax* DNA levels were determined by RT-PCR. Each dot represents one mouse. Shown in mean ± SD. Statistics: one-tailed Mann-Whitney test.

## Structural studies defining humAb 826827 interaction with PvAMA1

We obtained a 2.4 Å resolution structure of humAb 826827 bound to the PvAMA1 ectodomain with an $R_{work}/R_{free}$ of 19.2% and 23.5% (Supplemental Table 4). A 1:1 complex is present in the asymmetric unit (Fig. 5A). The resulting electron density map allowed the tracing of PvAMA1 from residues 46 to 474 with main chain gaps at residues 211−215, 296−303, 328−334, and 402−415. Heavy and light chains of the antigen-binding fragment of 826827 were fully traced from 1−231 and 1−214, respectively. Our structure shows that 826827 interacts with PvAMA1 residues of Domain 1 and the mobile loop of Domain 2 (Fig. 5A, **panel i and ii**). Five of the six complementarity-determining regions (CDR, namely: L1, L2, H1, H2, and H3) form direct contacts with PvAMA1 with a buried interaction surface of ~1392 Å², with the CDR-H3 loop of 826827 contributing 70% of the buried surface area (Fig. 5B, Supplemental Figure 10). The CDR-H3 loop of 826827 forms a disulfide bridged β-hairpin that binds to the hydrophobic groove on PvAMA1 Domain 1, which constitutes part of the RON2-loop receptor binding site (Fig. 5A−C). CDR-H3 binding to the PvAMA1-hydrophobic groove involves 53 interatomic contacts with distances <3.8 Å of which six are hydrogen bonds (Supplemental Table 5). The mobile Domain 2 loop of PvAMA1 is contacted by residues of CDR-H3 and CDR-H1, forming one salt bridge and six hydrogen bonds that stabilize its position on Domain 1 (Fig. 5, Supplemental Fig. 11). CDR-L2 and H2 interact with PvAMA1 residues of Domain 1 loops that surround the mobile Domain 2 loop, and CDR-L1 forms contacts with a Domain 1 loop next to the hydrophobic groove (Fig. 5). Two residues located in the Domain 2 loop of AMA1, Arg317 and Lys321, provide a positively charged patch at the bottom of the RON2-loop binding groove. Otherwise, this binding pocket is largely hydrophobic (Supplemental Fig. 7). One salt bridge between PvAMA1 Lys321 and humAb 826827 CDR-H3 Glu103 provides an anchoring and orientation point for the observed interaction between the two proteins. The remainder of the interactions are hydrophobic and hydrogen bonding interactions in nature (Fig. 5i, Supplemental Movie 1, Supplemental Movie 2).

Compared to structures of unbound PvAMA1 (PDB ID: 1W8K[42]) and the PvAMA1 bound to a peptide representing the ß-hairpin loop of RON2 (PDB ID 5NQG[26]), our PvAMA1 structure overlays with low root mean square (r.m.s.) deviation values of 0.463 Å (over 2137 atoms) and 0.417 Å (over 2121 atoms) respectively (Fig. 5D, E). A major difference exists in our structure, where the mobile Domain 2 loop with residues 304−327 is visible while it is unstructured in the other PvAMA1 structures. Structural analysis with *Plasmodium* AMA1 homologs showed that the Domain 2 loop of our structure adopts a similar 'closed' conformation as in unbound PkAMA1 (PDB ID: 4UV6[43]) which shares 90% sequence identity in this loop (Fig. 5F, G). While Akter et al.[55] observed an open or semi-open conformation in their PfAMA1 structure complex with a cyclized RON2 peptide (Supplemental Movie 3), our crystal structure clearly shows that in the presence of 826827, the Domain 2 loop of PvAMA1 is stabilized and remains bound to Domain 1, effectively blocking the RON2 binding site. The displacement of the Domain 2 loop is required to expose the complete RON2 binding site (Fig. 5F, G). Our proposed model aligns with molecular dynamics simulations performed on PfAMA1-PfRON2 and TgAMA1-TgRON2 complexes[56,57]. Our co-crystallization structure confirms that humAb 826827 binds to an important region of PvAMA1, providing a mechanistic explanation for its high potency.

## HumAb 826827 binds in a highly conserved binding pocket of PvAMA1

Previous co-crystal structures of Pf- and PvAMA1 reveal a hydrophobic binding pocket on AMA1 that interacts with the extracellular RON2-loop[23,26,45,58]. Visually, our co-crystallization structure of humAb 826827 with PvAMA1 indicates that this antibody may interact with the same PvAMA1 residues as the RON2-loop. Figure 6A shows that the contact residues between PvAMA1 and RON2-loop overlap with those of PvAMA1 that contact humAb 826827 CDR-H3 residues. Sequence and structural analysis of the contact residues indicate that humAb 826827's epitope is highly conserved for PkAMA1 and PcAMA1. However, little epitope conservation is observed in PfAMA1, accounting for the lack of humAb recognition of the recombinant protein (Figs. 1C, 6A; Supplemental Fig. 6). Comparing the RON2 and CDR-H3 binding site of 826827 to other available *Plasmodium* species and model systems reveals that *P. cynomolgi* AMA1 is 100% conserved and would therefore serve as a predictive non-human primate model for Pv challenge infections to evaluate humAb 826827 (Fig. 6A, Supplemental Fig. 8).

## HumAb 826827 competes for the same epitope on PvAMA1 as PvRON2

To confirm that the binding of humAb 826827 to PvAMA1 inhibits the extracellular RON2-loop from binding, we performed a dose-response competition assay between the two proteins. We found that humAb 826827 at a concentration as low as 2.5 μg/mL inhibits the binding of the biotinylated PvRON2-loop peptide to PvAMA1 (Fig. 6B). This result demonstrates that humAb 826827 competes for the same epitope as

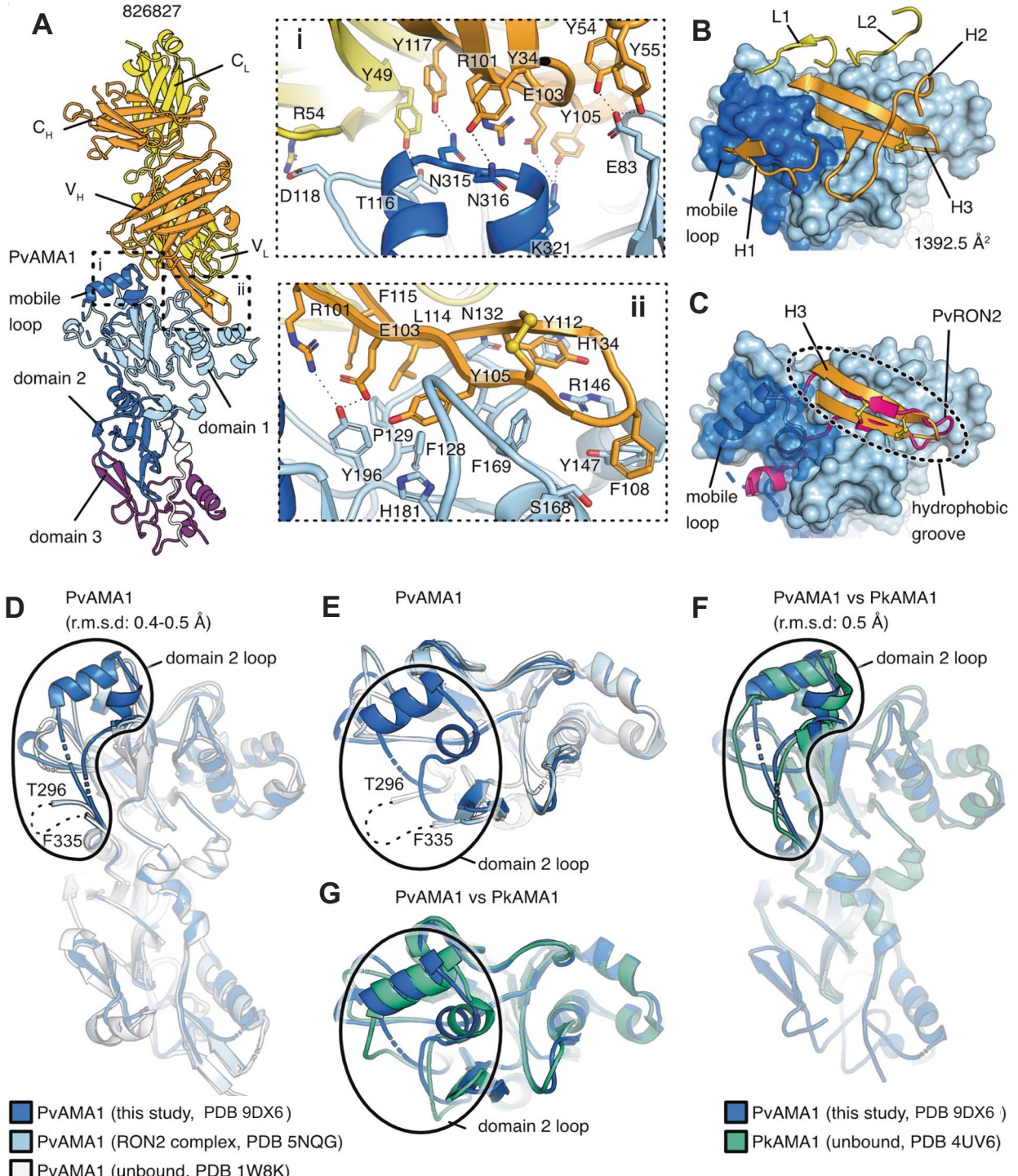

**Fig. 5 | Crystal structure of the PvAMA1–Fab 826827 complex. A** Ribbon representation of PvAMA1–Fab 826827 complex (PDB ID: 9DX6). The heavy chain of 826827 is orange, the light chain is yellow. PvAMA1 with N-terminal extension (residues 46-62) is white, Domain 1 (residues 63-248) light blue, Domain 2 (residues 249-385) dark blue, and Domain 3 purple (residues 386-474)[51]. Close-up views show interactions between humAb 826827 and (**panel i**) the mobile Domain 2 loop of PvAMA1 and (**panel ii**) the hydrophobic groove. For clarity, only polar interactions between side chains are indicated with dotted lines. **B** Five CDR loops (L1, L2, H1, H2, and H3) are involved in PvAMA1 binding. PvAMA1, in surface representation, is colored as described in panel A. The size of the buried interaction surface is indicated. **C** Comparison of PvAMA1 when bound by CDR-H3 of Fab 826827 or PvRON2 peptide (PDB ID: 5NQG). PvAMA1 is shown with transparency around the Domain 2 loop (dark blue). CDR-H3 and PvRON2 peptide (magenta) form disulfide-linked β-hairpin loops that bind to the hydrophobic groove on Domain 1. HumAb 826827

stabilizes the Domain 2 loop in a closed position on Domain 1. This loop is dislocated when PvRON2 peptide is bound to PvAMA1. **D** Ribbon representation of overlaid PvAMA1 structures, comprising our structure of PvAMA1 in complex with Fab 826827 (blue, PDB ID: 9DX6), PvAMA1 in complex with PvRON2 peptide (light blue, PDB ID: 5NQG), and unbound PvAMA1 (gray, PDB ID: 1W8K). This indicates that 826827 stabilizes the Domain 2 loop and allows for the refinement shown. Residues between T296 and F335 are absent in the electron density map of PDB 5NQG and 1W8K but residues 304−327 are well defined in our structure. Overlays were generated using Pymol, and refined root mean square deviation values are indicated. **E** Top view of the PvAMA1 overlays. **F** Overlay of PvAMA1 (blue, PDB ID: 9DX6) with PkAMA1 (green, PDB ID: 4UV6). The mobile Domain 2 loops adopt a similar conformation in both structures. **G** Top view of PvAMA1 and PkAMA1 overlay. (see **Supplemental** Table 4 for data collection and refinement statistics).

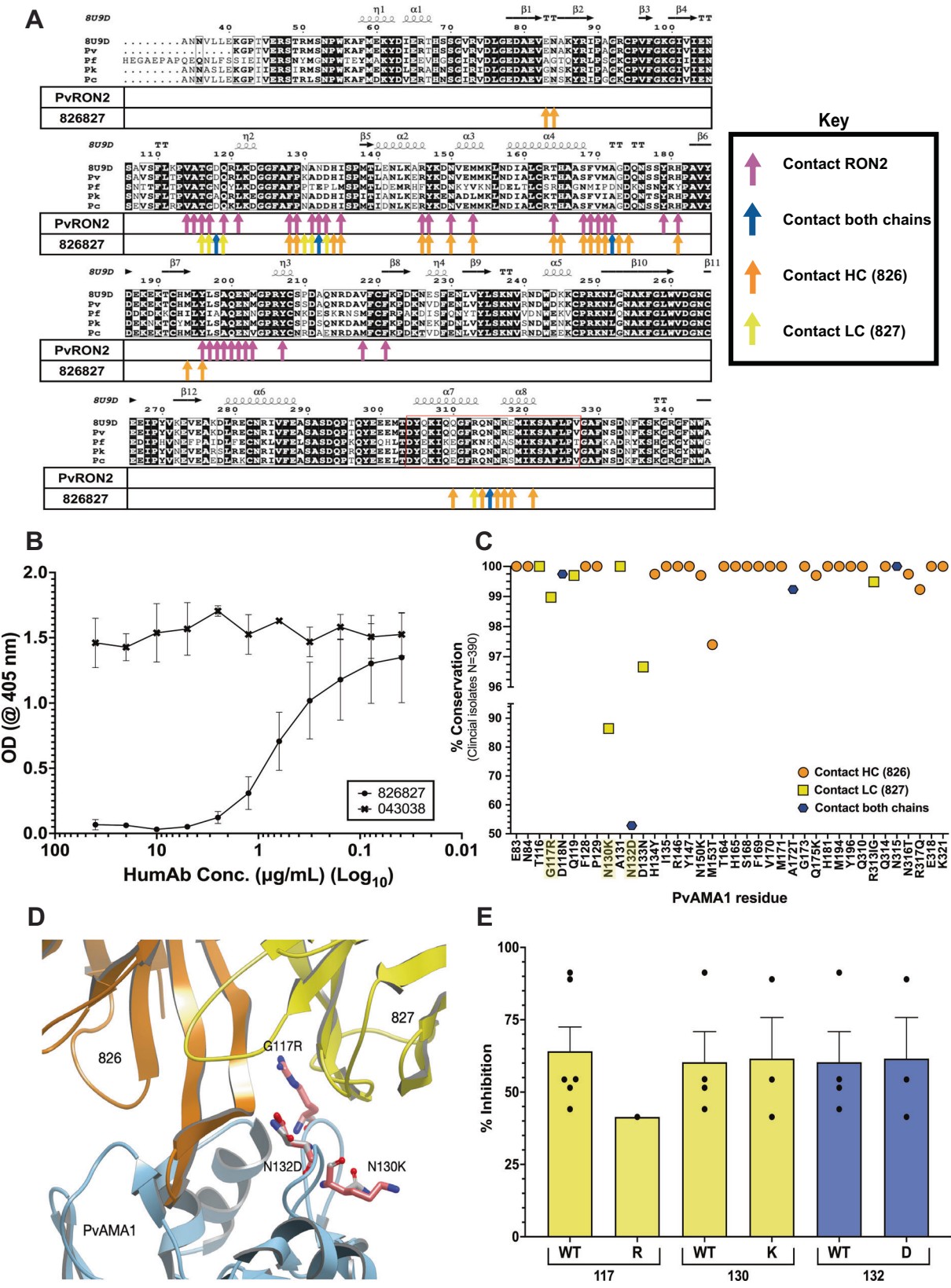

the RON2-loop, further confirming humAb 826827-induced invasion inhibition mechanism.

## Sequence conservation across 390 clinical PvAMA1 isolates

To evaluate how the sequence conservation and polymorphisms in PvAMA1 may impact humAb efficacy, we examined 390 published PvAMA1 sequences from clinical isolates. We found 98%

conservation of PvAMA1's total amino acid sequence across the 484 residues. Domain 1 is the largest and most polymorphic of the domains of PvAMA1[51]. Domain 1 (1–248) contains 21 polymorphic amino acid residues, Domain 2 (248–385) contains 6, and Domain 3 (386–484) contains 3. All polymorphic residues are 44–80% conserved across 390 PvAMA1 sequences (Supplemental Fig. 9[51]).

**Fig. 6 | Interaction residues of PvAMA1 with RON2 and 826827. A** Sequence alignment of published AMA1 amino acid sequences with the corresponding binding residues for RON2 (magenta) and humAb heavy chain (826; orange), light chain (827; yellow), or both chains (blue) arrows. The first line of the alignment represents the deposited structure 9DX6 corresponding to the PvAMA1 Palo Alto strain (ACB42438). Pv is the Sal1 variant (PVX_092275). The red box outlines the Domain 2 loop (304–327), highlighting 826827's interactions with this region of PvAMA1. **B** Competition assay using 50 µg/mL of PvRON2 (Asp 2050 – Thr 2088) that competed with varying concentrations of humAb 826827 (40–0.039 µg/mL) to bind recombinant PvAMA1. 043038 was used as a negative control. Error bars indicate +/− SEM. **C** PvAMA1 residues contacting humAb heavy chain 826 (orange

circles), light chain 827 (yellow squares), or both humAb chains (dark blue hexagons) and their mutations are displayed on the X-axis while the Y-axis shows the conservation of that position within 390 clinical isolate sequences. Highlighted amino acids represent those depicted in Fig. 6D. **D** Structural analysis of the interaction between 826 (orange) 827 (yellow) and PvAMA1 (blue) shows three observed polymorphisms in the binding epitope (G117R, N132D, and N130K; pink). **E** The *pvama1* gene was sequenced in the seven Pv clinical isolates (**Supplemental Table 3**). Amino acids 117, 130, and 132 represent polymorphic contact residues of humAb 826827 to PvAMA1 Sal1 reference strain (PVX_092275). Bars represent the mean ( + SEM) of humAb invasion inhibition of WT compared to an isolate expressing a SNP at each of the three polymorphic residues ($n = 7$).

There is 97% conservation of PvAMA1 residues that contact the CDR3 loop of 826827 (Fig. 6C). We evaluated whether the PvAMA1 polymorphisms of the Pv clinical isolates used in the invasion assays affected the potency of 826827. We classified the invasion data by assessing the impact of the polymorphisms of AMA1 within the epitope recognized by humAb 826827 (Supplemental Table 3). None of the 7 isolates tested against 826827 (Fig. 2B) shared the same haplotype with the wild type (WT) reference Sal1 strain (PVX_092275) and had unique amino acid sequences. Among the 7 isolates, 4 were WT for the residues recognized by 826827; 2 had two mutations, and 1 had three mutations. All mutations were observed in residues 117, 130, and 132. As predicted, these polymorphisms did not impact the invasion inhibition by 826827 in terms of number of mutations compared to the WT or when considering individually each mutation compared to the WT, supporting our structural interpretation of the interaction (Fig. 6D, E; Supplemental Tables 3, 5).

Structural analysis of the two residues N130 (N 86.4%, K 13.9%) and N132 (D 52.9%, N 46.7%, G 0.5%) with the highest variability between PvAMA1 sequences are unlikely to disrupt humAb 826827 binding significantly as the interactions are either maintained (backbone interactions) or do not directly lie in the groove of the binding pocket (Fig. 6D). An additional mutation within hydrogen bonding distance in the proximity of the CDR3 loop is G117 (G 99.0%, R 1%), which is unlikely to alter the interaction as it is pointing towards the solvent and is not interacting with other residues on the protein (Fig. 6D; Supplemental Movie 2). These data support our observations that humAb 826827 consistently inhibit liver and reticulocyte invasion with multiple clinical isolates.

## Discussion

We describe the first reported humAb that recognizes PvAMA1 and inhibits the invasion of pre-erythrocytic and blood-stage parasites. HumAb 826827 is potent and blocks the invasion of multiple clinical isolates. The dual activity against sporozoites and merozoites is valuable because new infections and relapses from dormant liver hypnozoites drive Pv disease[59]. Reducing blood-stage infection also attenuates gametocyte production and, thus, Pv transmission[60].

826827 was the most potent humAb to inhibit merozoite invasion into erythrocytes using a Pf transgenic parasite line expressing PvAMA1 (IC$_{50}$ = 3.0 µg/mL)[52] and in short-term invasion experiments into reticulocytes using Pv clinical isolates (IC$_{50}$ = 48 µg/mL). Only humAb 826827 showed significant inhibition of multiple Pv clinical isolates consistent with its recognition of a conserved epitope on PvAMA1. The other growth-inhibiting humAbs may target polymorphic epitopes that differ among the clinical isolates from the PvAMA1_Palo Alto, the variant used in Pf transgenic parasites and to sort B cells. These humAbs are more potent than previously reported PfAMA1-specific murine mAbs IF9 and 4G2, which display IC$_{50}$s of 292 and 105 µg/mL, respectively, against the Pf WT strain 3D7 in blood stage invasion studies[24,45,50,61,62] and rat mAb R31C2[43,58]. They exhibit similar potency to a PfAMA1 humAb produced from an IgG sequence isolated from a Ghanaian with an IC$_{50}$ of 35 µg/mL against the Pf 3D7 variant

in vitro[50]. Of note, a single-component PfAMA1-RON2L immunogen was developed as a vaccine candidate to produce an antibody response to complexed PfAMA1-RON2. Although the elicited polyclonal antibodies displayed Pf strain-transcending properties like our humAb 826827, they had poor potency with IC$_{50}$s ranging from 1.5 to 4.5 mg/mL against Pf 3D7[63].

HumAb 826827 also blocked sporozoite invasion into human hepatocytes. This is consistent with recent studies showing AMA1 is utilized during sporozoite penetration of hepatocytes and entry into mosquito salivary glands[16,17]. It has been previously demonstrated that mouse polyclonal antibodies generated by PfAMA1 or PvAMA1 inhibit blood sporozoite invasion into human hepatocytes but require a concentration of 0.5 to 1 mg/mL[17]. By contrast, our humAb 826827 has an IC$_{50}$ of 0.3 µg/mL when preventing sporozoite invasion of human hepatocyte cell line HC04 and an IC$_{50}$ of 3.7 µg/mL in primary human hepatocytes. Using different clinical isolates across the different assays and observing slight variation in inhibition suggests that humAb 826827 is strain-transcendent with high potency. Indeed, the potency of humAb 826827 was comparable to the murine anti-circumsporozoite protein mAb 2F2[53], which was used as a positive control in these experiments. This is notable as passive transfusion of a suboptimal dose of mAb 2F2 in liver-humanized mice and challenged with Pv sporozoites reduced parasite relapse by 62%, associated with a corresponding reduction of hypnozoite numbers[12]. This suggests that humAb 826827, when used in that same model, could achieve a similar or more significant reduction in hypnozoites, though this would require future testing. Of note, we validated humAb 826827 ability to inhibit sporozoite invasion into primary human hepatocytes. This model yields much higher infection rates than the HC04 cell line and allows for the development of both hypnozoites and schizonts[64].

The difference in humAb 826827 concentrations required to inhibit sporozoites (IC$_{50}$ = 0.3–3.7 µg/mL) in hepatocytes and merozoite invasion (IC$_{50}$ = 48 µg/mL) may be attributed to variations in levels of antigen availability or the nature of molecular interactions that occur during invasion. Parasite load, and thus, the number of PvAMA1 molecules expressed, differs significantly between the different in vitro assays. While $1 \times 10^5$ sporozoites are used in the liver invasion assay, the number of merozoites present during the clinical isolate blood stage assay is a log-fold higher. Furthermore, the structural conformation of different epitopes displayed may vary between pre-erythrocytic and erythrocytic stages of invasion. However, the combination of our pre- and erythrocytic invasion inhibition data confirms that PvAMA1 plays an important role during sporozoite invasion into hepatocytes and merozoite invasion into reticulocytes. This confirms that AMA1 is a viable multi-stage therapeutic target against *Plasmodium* infection and requires further research.

The humAb 826827 also reduced liver-parasite burden in vivo using an FRG-humHep human liver chimeric mice model. When the animals received 300 µg of PvAMA1-specific humAb 826827 and achieved mean blood antibody levels of 27.9 µg/mL two days after sporozoite challenge, there was a significant decrease in liver parasite burden. This confirmed the in vitro observations. Administering a

lower dose (30 μg) of humAb 826827 resulted in high variability in liver infection. Some animals had low parasite burdens in the liver comparable to those observed with higher antibody doses. The variation in liver burden did not correlate with serum antibody levels at the lower humAb dose. However, the variability may be attributed to the difference in the viability of injected sporozoites or variation in the AMA1 expression on the sporozoites that could affect antibody efficacy at lower doses. There are important limitations to this model. The animals are immune deficient, and human mAb was used; both may impair or prevent activation of complement or antibody-dependent parasite elimination, which is known to be an important mechanism for parasite elimination[65]. This model only tested activity against sporozoite invasion of the liver, not the blood-stage infection. Intravenous injection of parasites circumvents sporozoite migration in the skin, which might also be susceptible to antibody elimination. However, FRG-humHep chimeric mice are a recognized screening tool that allowed us to evaluate the inhibitory potential of humAb 826827 in vivo.

The crystal structure of recombinant PvAMA1_Palo Alto bound to humAb 826827 reveals why this humAb is potent and strain transcendent. The CDR3 of the heavy chain (826) recognizes a conformational epitope that overlaps with the RON2-loop binding site in Domain 1 and displays a higher affinity for PvAMA1 than PvRON2, 30.5 nM and 50.0 nM respectively. Previous structural studies suggested that a mobile loop of PvAMA1 Domain 2 partially obstructs the RON2 binding grove. This mobile loop must be displaced for a successful PvAMA1:RON2 interaction to occur[18,43,44]. HumAb 826827 binds the Domain 2 loop, thus preventing displacement and further interfering with PvRON2 engagement of PvAMA1. The PvAMA1 contact residues that directly interact with humAb 826827 are conserved or possess single nucleotide polymorphisms (SNP) that do not affect humAb potency. For example, the SNP (N132D) frequently occurs in Pv clinical isolates. The Pv clinical isolates data indicates this mutation does not impair the hydrogen bonding capabilities between PvAMA1 and humAb 826827 (Fig. 6D). Similarly, D133N maintains hydrogen bonding capabilities and does not impact the $IC_{50}$ of 826827 (Fig. 6D). The mutation N130K does change the interaction between PvAMA1 and 826827. However, the sidechains point towards the solvent, and only the backbone of the residue interacts with the light chain of 826827. As expected, there is no variation in humAb activity (Fig. 6D). Of note, 826827 displays a much higher avidity to PkAMA1 than to PvAMA1_Palo Alto. Two amino acid changes (M153L & M171I) occur in 826827's binding epitope from the PvAMA1 to PkAMA1 amino acid sequences. This variation in amino acid sequence creates a more hydrophobic and presumably stronger interaction between 826827 and PkAMA1, likely leading to increased avidity.

Recent studies suggest that PvAMA1 might interact directly with reticulocytes independent of RON2[28,29]. Another study shows that some antibodies that do not inhibit PfAMA1-PfRON2 interaction can still be protective[63]. This suggests that there might be antibodies that block AMA1 action by not blocking RON2. We did not find such antibodies to PvAMA1. The humAbs to PvAMA1 that blocked sporozoite or merozoite invasion also inhibited PvAMA1-RON2 interaction to some extent. We may have biased selection of PvAMA1-RON2 blocking Abs because the individual from whom we isolated humAbs had potent PvAMA1-RON2 blocking activity in serum. The isolation of humAbs from PvAMA1 by other individuals might identify such humAbs.

In conclusion, we have discovered a highly conserved epitope of PvAMA1 that can be targeted by humAb 826827, preventing AMA1-dependent sporozoite invasion into hepatocytes in vitro and in vivo and merozoite invasion into reticulocytes. HumAb 826827 may lead to developing a new treatment to combat Pv infection, disease, and transmission. Moreover, identifying a conserved inhibitory epitope of PvAMA1 effectively targeted by 826827 could guide the design of structure-based vaccines for Pv. Future research involving non-human

primate models could yield valuable data to support the development of humAb 826827 as a therapeutic or prophylactic agent.

## Methods

### Blood samples
Peripheral blood mononuclear cells (PBMC) were obtained from Cambodians with documented Pv malaria residing in Pursat Province, as described previously[49]. Samples were screened for blocking antibodies to PvAMA1 (see below). Institutional review boards from the National Institutes of Health (National Institute of Allergy and Infectious Diseases protocol 08-N094, clinicaltrials.gov identifier NCT00663546), Cambodian Ministry of Health, and University Hospitals of Cleveland Medical Center Institutional Review Board (no. 04-14-19), approved the protocols. Written informed consent was obtained from all study participants or their parents/guardians.

### Protein expression and purification
The protein sequences for each of the antigens were selected: PfAMA1 (XP_001348015.1; 3D7 genotype), PvAMA1 (ACB42433.1; Palo Alto genotype), PvAMA1 (PNG 20 genotype) and PkAMA1 (XP_002259339.1; Strain H) as previously described[29]. At the N-terminus of the sequences, the native signal peptide was removed and replaced with a signal peptide for tissue plasminogen activator, followed by a 6-histidine tag. At the C-terminus of all sequences, the sequence was truncated to remove the transmembrane domain and cytoplasmic tail. All sequences were assessed for potential glycosylation sites (https://services.healthtech.dtu.dk/services/NetNGlyc-1.0/). The AMA1 sequences for each species were then modified to prevent potential glycosylation (PfAMA1, six changes; PvAMA1, three changes; PkAMA1, seven changes)[29].

### Preparation of the PvAMA1 probe and cell sorting
The addition of biotin to PvAMA1_Palo Alto was done using EZ-Link NHS-PEG12-Biotin based on the manufacturers' instructions. Using a Zeba spin 7 kDa cutoff buffer exchange spin column eliminated free biotin and changed to Bicine buffer, 50 mM pH 8.3. PvAMA1_Palo Alto biotin was quantified by BCA Protein assay (Pierce) and biotin/molecule was calculated as 2 biotin per molecule using FluoReporter® Biotin Quantitation Assay, Invitrogen.

CD19+ human B cells were isolated from $30 \times 10^6$ cryopreserved PBMCs using Miltenyi CD19 microbeads and then stained at 4 °C for LIVE/DEAD, CD20, IgG, PvAMA1_Palo Alto_biotin tetramers prepared with Streptavidin-FITC and Streptavidin-Brilliant Violet 421. Single cells were sorted as a 96 well sort to a chilled wells on a FACS_ARIA_SORP. 252 single cells were isolated into 4 μL of catch buffer. Stained CD19 cells were stored at 4 °C overnight and 252 more cells were collected as a dry catch.

### Cell staining and sorting of PvAMA1-specific Memory B cells
Single cells were identified and sorted according to previously described techniques[66] from cryopreserved PBMC without activation. Briefly, staining and single-cell sorting of PvAMA1-specific IgG+ MBCs were performed as follows: B cells were enriched using immunomagnetic positive selection with anti-CD19 magnetic MACS beads (Miltenyi Biotec) and stained with mouse anti-human CD20 (PE-Cy5.5; Invitrogen) and anti-human IgG Abs (PE-Cy7 clone G18-145; Becton Dickinson) along with biotinylated PvAMA1 using Streptavidin coupled with allophycocyanin (Becton Dickinson) and SYTOX Green Dead Cell Stain (Invitrogen) to gate out dead cells. Stained CD19$^+$ cells were sorted on a BD FACSAria II equipped with chilled stage sorting based on size and complexity. Doublet discrimination was performed to exclude aggregated cells. Individual AMA1-specific CD20$^+$, IgG+ MBCs were single cell sorted directly into 4 μL of mRNA extraction buffer on a cooled 96-well metal block. After cells were collected, plates were frozen immediately on dry ice and stored at −80 °C until further processing.

**Table 1 | PCR Round 1 Master Mix Specifics**

| Component | Volume for 1 well (μL) |
|---|---|
| RNAse Free Water | 10.79 |
| 5x Q-Solution | 4.0 |
| 10x HotStar Taq PCR Buffer | 2.0 |
| 25 mM MgCl2 | 0.55 |
| dNTP Mix (25 mM each dNTP) | 0.20 |
| 5′ LV-Primer Group Mix | 0.130 |
| 3′ Primer (50 mM) | 0.130 |
| HotStar Taq (5 units/μL) | 0.20 |
| DNA Template | 2.00 |
| Total Volume (w/out DNA) | 18.00 |

**Table 2 | PCR Round 1 Primers**

| | 5′ | 3′ |
|---|---|---|
| Gamma | LVH Group Mix | CGammaCH1 |
| Kappa | LVKappa Group Mix | CK543 |
| Lambda | LVLambda Group Mix | Clambda |

**Table 3 | PCR Round 2 Primers**

| | 5′ | 3′ |
|---|---|---|
| Gamma | AgeVH Group Mix | IgGInternal |
| Kappa | PanVK | CK494 |
| Lambda | AgeVLambda Group Mix | CLambdaXho |

**Table 4 | Specific V(D)J Region Amplification Master Mix Specifics**

| Component | Volume for 1 well (μL) |
|---|---|
| RNAse Free Water | 18.0 |
| 5x Q-Solution | 8.0 |
| 10x HotStar Taq PCR Buffer | 4.0 |
| 25 mM MgCl2 | 1.0 |
| dNTP Mix (25 mM each dNTP) | 0.4 |
| 5′ primer at 5 μm | 4.0 |
| 3′ primer at 5 μm | 4.0 |
| HotStar Taq (5 units/μL) | 0.4 |
| DNA Template | 4.2 |
| Total Volume (w/out DNA) | 39.8 |

## cDNA synthesis

The 96-well plates with single cells were thawed on ice; a cold volume of 7 μL containing 300 ng of random hexamers (Qiagen Operon), 12 U Rnasin (Promega), and 0.9% NP-40 (Thermo Scientific Pierce) was added to each well. After thorough pipetting and rinsing, wells were capped, centrifuged at 4 °C, heated to 68 °C in a thermal cycler for 5 min, and placed on ice for at least 1 min. Reverse transcription was performed with the addition of 7 μL containing 3.6 μL of 53 reverse transcriptase buffer, 10 U RNAsin (Promega), 62 U Superscript III reverse transcriptase (Invitrogen), 0.62 μL dNTPs 25 mM each (V Bio-Tek), and 1.25 μL of 0.1 M DTT (Sigma-Aldrich). All wells were capped, and the plate placed in a cold rack and vortexed for 10 sec before centrifugation at 300 x $g$. Thermal cycler conditions for reverse transcription were as follows: 42 °C for 5 min, 25 °C for 10 min, 50 °C for 60 min, 94 °C for 5 min, and 4 °C hold. When completed, 10 μL of nuclease-free PCR water was added to each well.

## Ig gene amplification

Immediately following cDNA synthesis, IgG genes (Igg) were amplified in a total of 20 μL per well for the first round of nested PCR for IgG H chain (Iggh), IgG k (Iggk) and IgG l (Iggl), using master mix (Table 1) and primers (Table 2) as previously described REF. cDNA from individual sorted B cells were added and Igg amplified under the following conditions: thermal cycle PCR at 94 °C for 15 min; 50 cycles at 94 °C for 30 sec, then 58 °C (Iggh and Iggk) or 60 °C (Iggl) for 30 sec, then 72 °C for 55 sec; then one cycle at 72 °C for 10 min. Second round of nested PCR for Iggh, Iggk, and Iggl used master mix (Table 1), 2 μL of first-round PCR product with second-round primers (Table 3) and the same master mix protocol, with the following conditions; thermal cycle PCR at 94 °C for 15 min; 50 cycles at 94 °C for 30 sec, then 58 °C (Iggh and Iggk) or 60 °C (Iggl) for 30 sec, then 72 °C for 45 sec; then one cycle at 72 °C for 10 min. The PCR product generated was purified and sequenced, with V(D)J genes determined using IMGT/V-Quest.

## Specific V(D)J region amplification and cloning

Primers specific with restriction enzyme sites for V and J regions were used to amplify the first-round PCR product to generate a fragment for cloning based on previously described primers[66]. PCR product was purified, and restriction enzyme digested, cloned into Iggh, Iggk, or Iggl expression vectors, and chemically transformed into 5 μL of

aliquots of TOP10 *E. coli* cells (Thermo Fisher Scientific). Successful transformants were screened by PCR amplification (Table 4) using a vector-specific primer paired with an insert-specific primer with the following conditions; thermal cycle PCR at 94 °C for 15 min; 50 cycles at 94 °C for 30 sec, then 57 °C (Mu/Gamma/Kappa) or 60 °C (Lambda) for 30 sec, then 72 °C for 45 sec; then one cycle at 72 °C for 10 min. Products were sequenced and compared with the second-round PCR product sequence.

## Definition of clonal groups

Clonal groups were based on H-chain nucleotide sequences. Any PCR product with 0.8% nucleotide sequences with a Phred score of <20 was excluded. We determined H chain alleles from PCR-amplified sequences using IMGT/V-QUEST (http://www.imgt.org, international ImMunoGeneTic information system). Because of primer mixture ambiguities, the first 20-22 nt of IgGH variable regions were designated germline; thus, this region was not evaluated for somatic hypermutations. IMGT/V-QUEST was used to assign V(D)J organization and shared IgHV genes and CDR3 length-grouped sequences. Clonal grouping was determined using Sequence Manipulation Suite: Ident and Sim, and the AB model in the open-source software CodonPhyML described by A. Mirsky, et al.[67]. This software utilizes specific clusters TIVLMFA, YHWGCPQ, SNKR, and DE to calculate the similarity between CDR3 sequences. A clonal group is defined by the same VDJ gene usage, CDR3 length, 84% similarity CDR3 aa sequence.

## HumAb expression and purification

Briefly, following transformation and sequencing to confirm that the plasmid contains the desired nucleotide sequence, glycerol stocks of *E. coli* containing the proper plasmid were stored at −80 °C. Plasmids were harvested from ampicillin grown *E. coli* cultures (50 − 250 mL) using the ZymoPURE Midiprep Purification Kit. Purified DNA was quantified using a NanoDrop and stored at 4 °C until use. Transfection sizes were then dependent on DNA yield. Cells were transfected at a density of $3 − 5 × 10^6$ viable cells per mL. On the day before transfection (Day −1), Expi293F™ culture was split to a final density of $2.5 − 3.0 × 10^6$ viable cells/mL and allowed to grow overnight. The cells reached a density of approximately $4.5 − 5.5 × 10^6$ viable cells/mL at the day of transfection. Cells were diluted with Expi293™ Expression Medium, pre-warmed to 37 °C to a final density of $3 × 10^6$ viable cells/mL. Plasmid

DNA was diluted in Opti-MEM™ I Reduced Serum Medium at 1 μg/mL of transfected cells. ExpiFectamine™ 293 Reagent was diluted with Opti-MEM™ I Reduced Serum Medium and incubated at room temperature for 5 minutes prior to initiating the plasmid DNA complexation reaction. Diluted ExpiFectamine™ 293 Reagent was added to diluted plasmid DNA and mixed gently by swirling. ExpiFectamine™ 293/plasmid DNA complexes were incubated at RT for 10–20 minutes and slowly transferred to the cell culture with continuous mixing. Cells were incubated at 37 °C with a humidified atmosphere of 8% $CO_2$ in the air with continuous orbital shaking. 18–22 hours post-transfection, ExpiFectamine™ 293 Transfection Enhancer 1 and ExpiFectamine™ 293 Transfection Enhancer 2 were added to the flask per company instruction and gently mixed. The flask was returned to the 37 °C incubator with a humidified atmosphere of 8% $CO_2$ with shaking. Six to seven days post transfections cells are centrifuged at 4000 x *g* for 20 minutes at RT. The supernatant is decanted and filtered through a 0.22 μm sterile filter and stored at 4 °C until purification. The volume of the cleared cell supernatant was supplemented with 1 M Tris pH 8.0 and 5 M NaCl solutions to a final concentration of 0.1 M Tris pH 8.0 and 0.5 M NaCl and loaded onto a 5 mL HiTrap MabSelect PrismA column. Once the complete sample was loaded onto the column, the unbound sample was washed out with 10 column volumes (CV) 1x PBS. Elution was carried out with 10 CV using a commercial IgG Elution buffer pH 2.5 (Thermo Fisher) into a collection tube containing 1 mL of 1 M Tris pH 8.0 for immediate neutralization. Following the elution, the column was washed with 10 CV of 50 mM NaOH and re-equilibrated to 1 x PBS pH 7.4. Eluted humAbs were dialyzed overnight in 1 x PBS at 4 °C. Protein concentrations were determined by Nanodrop (Thermo Fisher Scientific) using OD 1.3 for a 1 mg/mL concentration.

### Antibody titration curve (MagPix) methods

A total of 3.6 μl of 1 mg/mL PvAMA1_PaloAlto was conjugated to 300 μL of MagPix bead 19 following the manufacturer's protocol (Luminex, Corp, Austin, TX). 1 μl of 4 mg/mL PfAMA1_3D7 was conjugated to 200 μL of MagPix bead 20, 4 μl of 1 mg/mL PvAMA1_PNG16 was conjugated to 200 μL of MagPix bead 18, 4 μl of 1 mg/mL PkAMA1 was conjugated to 200 μL of MagPix bead 56, 4 μl of 1 mg/mL TgAMA1 was conjugated to 200 μL of MagPix bead 12. 50 μL of humAb at the following concentrations (1.0, 0.5, 0.25, 0.125, 0.0625, 0.03125, 0.015625, 0.0078125 μg/mL) were mixed with 50 μL of bead master mix (diluted 1:1000 in 1% BSA-TBST). Samples were incubated for 10 min at RT while shaking, and an additional 20 min while sitting. Plate was placed on a magnet for 5 min and excess liquid was removed. Plate was washed two times with 1x TBST for 2 min with shaking, followed by an additional 5 min on a magnet. Donkey anti-Human IgG (PE conjugated, Jackson ImmunoResearch) secondary antibody was diluted 1:400. 25 μL of secondary antibody was added to the plate and samples were incubated for 10 min at RT while shaking, and an additional 20 min while sitting. Plate was placed on a magnet for 5 min and excess liquid was removed. Plate was washed two times with 1x TBST for 2 min with shaking, followed by an additional 5 min on a magnet. 150 μL of 1% BSA-TBST was added to the plate, shaken for 2 min, and read using the MagPix machine.

### Antibody avidity (MagPix) methods

The beads used in the humAb titration were also used to measure antibody avidity. 50 μL of 0.2 μg/mL of humAb was mixed with 50 μL of bead master mix (diluted 1:1000 in 1% BSA-TBST). Samples were incubated for 10 min at RT while shaking, and an additional 20 min while sitting. Plate was placed on a magnet for 5 min and excess liquid was removed. Plate was washed two times with 1x TBST for 2 min with shaking, followed by an additional 5 min on a magnet. 50 μL of Ammonium Thiocyanate at the following final concentrations of 0, 0.9, 1.2, 1.5, 1.8, 2.1, 2.4, 2.7, 3.0, 3.3, 3.6, 3.9 M was added to the plate.

Samples were incubated for 5 min at RT while shaking, and an additional 5 min while sitting. The plate was placed on a magnet for 5 min and excess liquid was removed. The plate was washed two times with 1x TBST for 2 min with shaking, followed by an additional 5 min on a magnet. Donkey anti-Human IgG (PE conjugated, Jackson ImmunoResearch) secondary antibody was diluted 1:400. 25 μL of secondary antibody was added to the plate and samples were incubated for 10 min at RT while shaking, and an additional 20 min while sitting. Plate was placed on a magnet for 5 min and excess liquid was removed. Plate was washed two times with 1x TBST for 2 min with shaking, followed by an additional 5 min on a magnet. 150 μL of 1% BSA-TBST was added to the plate, shaken for 2 min, and read using the MagPix machine. GraphPad Prism was used to calculate the $IC_{50}$ of each humAb.

### Measurement of antibody affinity

To determine the binding affinity of each humAb to PvAMA1, Surface Plasmon Resonance (SPR) was utilized to perform single cycle kinetics. One CM5 chip (GE Healthcare) flow cell was coated with 275 RUs of anti-Human IgG antibody (Abcam). An untreated cell was used as a blank reference. Data was collected at 25 °C and 10 Hz on a Biacore T200 system. Pilot studies of the interaction and follow-up regeneration protocols were established before data collection. Three start-up cycles were run with 1x HBSS Running Buffer (GE Healthcare). A humAb, at a concentration of 80 nM, was passed over all the flow cells at a rate of 10 μL/min for a contact time of 30 sec and allowed to stabilize for 120 sec. Increasing concentrations (5 nM, 10 nM, 20 nM, 40 nM, 80 nM) of rPvAMA1 were passed over all flow cells at a rate of 30 μL/min. Each injection was allowed to stabilize for 60 sec and was given 120 sec for dissociation. Anti-Human Fc chip was regenerated to baseline using IgG Elution Buffer (Thermo Scientific) and the process was repeated for all humAbs in the panel. Data was processed using the double referencing method using Scrubber 2.0 (BioLogic Software).

### Transgenic Pf-PvAMA1 growth inhibition assays

Invasion assays were performed as previously described[68], with minor changes. Parasites were synchronized with sorbitol, then inoculated at late trophozoite and schizont stages at 0.5% parasitemia in 4% hematocrit in 45 μL RPMI media, with 5 μL antibodies in 1x PBS. Cultures were left to invade, then harvested at the ring stage by fixation with 0.25% glutaraldehyde. Parasites were stored in 1x PBS at 4 °C. Cultures were later stained with 1× SYBR Green I (S7563, Invitrogen), and parasitemia assayed by flow cytometry in a BD FACSCanto™ II system, with further analysis in FlowJo™ v10.9.0 (BD Life Sciences). Parasites were gated from uninfected erythrocytes using the FITC-A and PE-A channels. Net growth rates were calculated by subtracting the parasitemia of parasites inoculated with 1 mg/mL heparin, then dividing by the parasitemia of parasites inoculated with 5 μL PBS. Growth percentages were subtracted from 100% to give invasion inhibition.

$IC_{50}$ calculations were performed in R [R Core Team (2023). R: A Language and Environment for Statistical Computing. R Foundation for Statistical Computing, Vienna, Austria. https://www.R-project.org/] using the packages drc[69] and ggplot2[70]. Assays were performed in biological triplicate, and $IC_{50}$s calculated independently for each replicate. The parasite lines were based on W2mef, with the AMA1 locus replaced with the *P. vivax* locus[52].

### Parasite culturing and transgenic PfPvAMA1 inhibition experiments

*P. falciparum* asexual stage parasites were maintained in culture in human erythrocytes (blood group type O +) at a hematocrit of 4% in RPMI-HEPES supplemented with 0.25% (w/v) Albumax™ (Invitrogen) and 5% (v/v) heat-inactivated human serum. *P. falciparum* was synchronized using sorbitol and heparin treatments as described previously[48]. Transgenic W2mef *P. falciparum* strains expressing

PfAMA1 W2mef allele (W2-W2), PfAMA1 3D7 allele (W2- 3D7) and PvAMA1 Palo Alto allele (W2-PvAMA1) were generated in previous studies[52].

### In vitro invasion inhibition assay using Pv clinical isolates, DNA extraction amplification and sequencing of PvAMA1

Clinical isolates of *P. vivax* were either used fresh after blood collection or cryopreserved and infected RBCs were thawed and cultured in IMDM medium (Gibco) supplemented with 0.5% Albumax II (Gibco), 2.5% heat-inactivated human serum, 25 mM HEPES (Gibco), 20 µg/mL gentamicin (Sigma) and 0.2 mM hypoxanthine (C-C Pro) for ~24 or ~48 hr until a majority of schizont stage parasites were observed as previously described[71]. The schizont-infected erythrocytes were enriched using KCl-Percoll density gradient[72], then mixed at a ratio of 1 erythrocyte to 1 reticulocyte enriched from cord blood and previously labeled with CellTrace Far Red dye following manufacturer's instructions. The cultures were incubated for ~10 hr in a final volume of 50 µL in 384 well plates, in presence of the humAbs. The threshold for *P. vivax* invasion of reticulocytes in controls is ≥0.2% parasitemia. This separates new invasion events from uninfected reticulocytes on FACS analysis. For the experiments performed with AMA1 humAbs, invasion rates in controls varied from 0.24 to 0.7%. The mouse monoclonal anti-Duffy 2C3 at 100 µg/mL was used as positive invasion inhibition control[71]. Cells were stained with DNA stain Höchst 33342 post-invasion and examined by flow cytometry. Reticulocytes, which were Höchst 33342 and Far Red positive, were scored as new invasion events. For quantification, data were normalized against parasites mock-treated with 1x PBS. Invasion of reticulocytes in absence of antibodies ranged from 0.2% to 12.2% (mean 3.37%). The work presented here was approved by the National Ethics Committee for Health Research of Cambodia (192NECHR). All patients and/or their parents/guardians provided informed written consent.

Genomic DNA was extracted from an aliquot of the parasitized blood samples using the QIAamp DNA Blood Minit Kit (QUIAGEN), according to manufacturer's instructions. PvAMA1 sequences were determined by PCR and Sanger sequencing (Macrogen, Seoul, South Korea) using the following conditions. The PCR was conducted in 20 µL reaction consisting of 2 µL of DNA, 0.25 µM of primers (Forward 5′-AAGCTGCTCACCCGTTAGTG-3′; Reverse 5′-GGGTGGGAAGGTG-CATTCTG-3′) and 1X HOT FirePol Blend MasterMix (Solis BioDyne, Tartu, Estonia) under the following conditions: 95 °C for 1 min, followed by 34 cycles 95 °C for 20 sec, 63.8 °C for 30 sec, 72 °C for 2 min and a final extension at 72 °C for 5 min. Nucleotides and corresponding amino acids were analyzed using MEGA 11 software. The sequences generated were compared with the reference strain Sal1 (PVX_092275).

### RON2 peptide competition

We conducted a RON2 binding assay as previously described[29] with a few modifications. 96−well plates were coated with PvAMA1. HumAb was serially diluted in 0.1% casein in 1x PBS by a factor of two, with concentrations ranging from 40−0.039 µg/mL. 50 µL of humAb was added to each well. 50 µL 0.1% casein in 1x PBS was added to additional wells as blank controls. The optical density (OD) of these were later subtracted from each of the other wells to correct for background. Each well, except for the blank control wells, was incubated with 50 µL of 1 µg/mL biotinylated PvRON2 peptide (Asp 2050 – Thr 2088)[29] in 1x PBS. Streptavidin−horseradish peroxidase conjugated protein (Thermo Fisher Scientific, 21130) was added at 1/500. Plates were incubated with ABTS for 20 min before reading.

### In vitro invasion into HC04 hepatocytes using Pv Sporozoite in Thailand and primary human hepatocytes in Cambodia

*Anopheles dirus* mosquitoes were fed on blood collected from symptomatic patients attending malaria clinics in Tak, Songkla, and Ubon-Ratchathani Provinces in Thailand and in Kampong Speu province in Cambodia. Samples were confirmed positive for only *P. vivax* via microscopy and RT-PCR. *P. vivax*-infected blood samples were collected from patients under the approved protocol by the Ethics Committee of the Faculty of Tropical Medicine, Mahidol University (MUTM 2018-016). Sample collection in Cambodia was approved by the National Ethics Committee for Health Research of Cambodia (192NECHR). Written informed consent was obtained prior to blood collection.

Briefly, *P. vivax* infected blood was drawn into heparinized tubes and kept at 37 °C until processing. Infected blood was washed once with RPMI 1640 incomplete medium. Packed infected blood was resuspended in warm, non-heat inactivated naive human AB serum (Thailand) or in heat-inactivated naive human AB serum (Cambodia) for a final hematocrit of 50%. Resuspended blood was fed to laboratory-reared female *Anopheles dirus* mosquitoes for 30 min via an artificial membrane attached to a water-jacketed glass feeder kept at 37 °C. Engorged mosquitoes were kept on 10% sugar at 26 °C under 80% humidity at the designated insectary at the Mahidol Vivax Research Unit or the Malaria Research Unit of Institut Pasteur du Cambodge. Sporozoites were dissected from the salivary glands of infected mosquitoes 14−21 days after blood feeding and pooled in DMEM supplemented with 200 mg/mL penicillin-streptomycin (Thailand) or in RPMI without NaHCO₃ (Cambodia)[8].

In assays performed in Thailand, a 96-well plate was seeded with HC04 hepatocytes (50,000 cells/well). Isolated sporozoites (100,000 per well) were pre-incubated for 30 min with the humAbs at various concentrations (0.1, 1, 10, 100, and 1000 µg/mL). Sporozoite/humAb mixture was added to hepatocyte culture and incubated for 4 days at 37 °C in duplicate. Murine anti-PvCSP polyclonal antibodies CSP 210 (clone 2F2) and 247 were used as positive controls[53]. Anti-tetanus toxoid humAb 043038 was used as a negative control.

In the assay performed in Cambodia a 384-well plate was seeded with primary human hepatocytes (18,000 cells/well from BioIVT M00995). Isolated sporozoites (16,000 per well) were pre-incubated for 20 min with the humAb 826827 at various concentrations (0.1, 1, 10, 100, and 1000 µg/mL), with the humAb 043038 at 1000 µg/mL and with non-treated cell culture medium (latter two as negative controls). Each condition was tested in five technical replicates. Cell culture medium was changed to non-treated medium after 24 h of infection and the plate was incubated for 6 days at 37 °C. Greater than 60 liver stage parasites/well in controls are used as cut-offs for a valid experiment. The mean of invasion controls varied from 72 to 904 for experiments using humAbs to PvAMA1. Cells were fixed and stained using sporozoite specific polyclonal antibodies UIS4 and DAPI. Wells were then analyzed using fluorescent microscopy and invaded hepatocytes were quantified.

### In vivo infection of mouse human liver chimera mice

All animal procedures were reviewed and approved by the Institutional Animal Care and Use Committee of the Oregon Health and Science University (IACUC protocol number: [IP00002518]). The FRG-humHep mouse studies were conducted similarly to studies previously published[73] with modifications. FRG-humHep mice (n = 12) on the C57Bl/6 background were purchased from Yecuris, Inc (Beaverton, OR, USA). Mice were pre-screened to have a serum human albumin level indicative of >90% humanization of hepatocytes. Mice were injected intravenously with the indicated quantity of antibody, diluted in PBS, approximately 3 hours before the challenge. *Anopheles dirus* mosquitoes were fed on blood collected from symptomatic patients of *Plasmodium vivax* in Thailand, parasites were allowed to mature to the sporozoite stage and migrate to salivary glands. Infected mosquitoes were shipped to the US. The salivary glands are dissected and diluted in Schneider's insect media to allow sporozoites released from salivary glands. Sporozoites were centrifuged at 400 x g supernatant to

remove salivary gland debris, re-washed, and counted. Mice were immediately injected with 400,000 freshly dissected sporozoites, diluted in Schneider's insect media.

Blood was collected on day -1 (pre-infection) on days 2 and 9 post-infection using heparinized capillary tubes and 100 µL of whole blood was transferred to 1.9 mL of nucliSENS Lysis Buffer (Biomerieux Inc. Cat# 200292). Blood was allowed to lyse at room temperature for at least 30 minutes before storage at −80 °C until qRT-PCR analysis. The remaining whole blood was centrifuged at 600 x $g$ for 10 minutes, and plasma was frozen to measure human mAbs levels. Mice were then humanely euthanized, and liver samples were collected. Liver pieces were weighed and placed in 4 mL of nucliSENS Lysis Buffer and subsequently homogenized using a Bead Ruptor Elite (OMNI International, model #: NE48611/A) with ceramic beads on the Liver-7mL tube setting 1 disruptor cycle at 5 m/s for 30 seconds. Following homogenization, the supernatant was collected for qRT-PCR quantification of parasite burden. The weight of the liver sample processed was used to standardize the resultant qRT-PCR values.

Liver and blood samples were blinded and sent to Dr. Sean Murphy's lab at the University of Washington for quantification of *Plasmodium* 18 s rRNA following published methods[74]. Another portion of the liver was sent to Dr. Ashley M. Vaughan's Lab at Seattle Children's Hospital for IFA analysis following published methods[75].

### Protein expression and purification of PvAMA1 and humAb 826827 for crystallization

For crystallization purposes, PvAMA1 (ACB42433.1; Palo Alto genotype[76]) and humAb 826827 were expressed using transient transfection of Expi293 cells at a density of $2.5 \times 10^6$ cells/mL following the manufacturer's protocol. Five days after transfection, supernatants were harvested by centrifugation at 7000 x g and dialyzed overnight into 30 mM Tris pH 7.5, 300 mM NaCl. Dialyzed PvAMA1-supernatant was incubated with Ni-NTA resin (Qiagen) for 1 hr at 4 °C followed by stepwise elution using gravity flow chromatography. Size exclusion chromatography (SD200 increase in 20 mM HEPES pH 7.5, 150 mM NaCl) was used as the final purification step. Supernatant of humAb 826827 was loaded onto a 5 mL Mabselect PrismA column (Cytiva) equilibrated in 1x PBS. After washing the column with 1x PBS, humAb 826827 was eluted using 0.1 M citric acid pH 3.0 and neutralized with 1 M Tris pH 9.0 and buffer exchanged into 1x PBS.

### Complex formation, crystallization, and structure determination

HumAb 826827 was incubated with enzyme FabALACTICA (IgdE) at a ratio of 50 µg enzyme/mg humAb for 24–48 hr at 37 °C. Fab fragments were purified by affinity chromatography using a 1 mL HiTrap Kappa-Select column (Cytiva), eluted with 0.1 M glycine pH 1.5 and immediately neutralized with 1 M Tris pH 9.0.

For crystallization, PvAMA1- Fab 826827 complexes were formed by incubating the PvAMA1 and Fab 826827 at a molar ratio of 1.2:1 for 1 hr on ice. This complex was separated from excess Fab using size exclusion chromatography in 20 mM HEPES pH 7.5, 150 mM NaCl. Crystallization screens were set up at the Monash Macromolecular Crystallization Platform (MMCP, Clayton, VIC, Australia) with 3 – 6 mg/mL at 20 °C. Initial PvAMA1-Fab 826827 crystals grew in 22% PEG Smear Broad, 0.1 M Tris pH 8.5 (BCS screen, Molecular Dimensions) and were optimized by additive screening in a sitting drop vapor diffusion experiment (Additive Screen HT, Hampton Research). A PvAMA1·Fab 826827 crystal obtained in 22% PEG Smear Broad, 0.1 M Tris pH 8.5, 0.3 M Glycyl-glycyl-glycine was harvested with 30% glycerol in mother liquor as cryo-protectant and used for data collection.

X-ray diffraction data was collected at the MXII beamline of the Australian Synchrotron at 100 K[77]. The XDS package[78] was used for data processing. Molecular replacement with Phaser was used to solve the phase problem using structural coordinates of PvAMA1 (PDB ID 5NQG[26]) and Fab coordinates of PBD ID 6FG1[79]. Iterative cycles of structure building and refinement was carried out using Coot[80] and Phenix[81]. Figures of the structure were prepared with PyMOL (Version 2.5.2 Schrödinger, LLC) or VIDA 4.4 (OpenEye, Cadence Molecular Science). The atomic coordinates and structure factor files have been deposited in the Protein Data Bank under PDB ID 9DX6.

### Reporting summary

Further information on research design is available in the Nature Portfolio Reporting Summary linked to this article.

## Data availability

Coordinates and structure factors of rPvAMA1_Palo Alto in complex with 826827 have been deposited in the Protein Data Bank (PDB) under PDB ID 9DX6. The data generated in this study for are provided in the Supplementary Information/Source Data file. Source data are provided with this paper.

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

## Acknowledgements

The work was funded by The National Institutes of Health (R01AI143694) and Veterans Affairs Research Service (I01 BX001350). JGB is supported by a NHMRC Investigator Grant (1173046). WHT is supported by a NHMRC Investigator Grant (2016908). Burnet Institute is supported by infrastructure grants from the NHMRC and Victorian State Government. We thank Damien Drew for the database analysis and for designing the PNG16 sequence for expression as a recombinant protein. We thank the individuals who donated their blood for this project and Monash Macromolecular Crystallization Platform (MMCP, Australia) for assistance with setting up the crystallization screens. This research used the MX2 beamline at the Australian Synchrotron, part of ANSTO, and used the Australian Cancer Research Foundation (ACRF) detector. We thank Martin Boulanger for recombinant TgAMA1. We thank the Cytometry & Imaging Microscopy Shared Resource of the Case Comprehensive Cancer Center (P30CA043703 and S10OD021559) for their support. The views are of the authors and not those of NIH and VA Research Service.

## Author contributions

J.B., C.L.K., J.G.B., W.H.T, and J.P. conceived and designed the experiments. L.L.C., A.N.M., K.R.R., O.M., and Y.S.-P. isolated and generated the humAbs. A.C.W., W.R., J.S., L.B.F.-D., L.B., C.D., A.O., and L.M.Y. performed the functional experiments. L.M.Y., D.H.O., J.G.B., J.B., C.L.K., J.P., A.O., and A.C.W. conducted data analysis and interpretation. L.M.Y., D.H.O., and J.G.B. generated and provided key reagents (recombinant proteins, antibodies, RON2 peptides, parasite lines). MHD and NCJ expressed and purified the recombinant proteins required for protein crystallography, performed crystallization experiments, and collected diffraction data at the Australian Synchrotron. M.H.D. determined the crystal structure. P.K., K.N., M.A., and B.K.W. performed the in vivo experiment and subsequent tissue collection. G.Z., N.R., and A.M.V. performed the IFAs and analysis. All authors contributed to the writing of manuscript and A.C.W. J.B., and C.L.K. completed the first draft of the manuscript. All authors have read and approved the final manuscript. The authors declare no competing interests.

## Competing interests

The authors declare no competing interests.

## Additional information

[1]Center for Global Health and Diseases, Department of Pathology, Case Western Reserve University School of Medicine, Cleveland, USA. [2]Walter and Eliza Hall Institute of Medical Research, Parkville, Victoria, Australia. [3]Department of Medical Biology, The University of Melbourne, Parkville, Victoria, Australia. [4]Burnet Institute, Melbourne, Victoria, Australia. [5]Department of Medicine, The University of Melbourne, Parkville, Victoria, Australia. [6]Mahidol Vivax Research Unit, Faculty of Tropical Medicine, Mahidol University, Bangkok, Thailand. [7]Department of Microbiology, Monash University, Clayton, Victoria, Australia. [8]Malaria Research Unit, Institut Pasteur du Cambodge, Phnom Penh, Cambodia. [9]Department of Infectious Diseases, The University of Melbourne, Parkville, Victoria, Australia. [10]Central Clinical School and Department of Microbiology, Monash University, Clayton, Victoria, Australia. [11]Vaccine and Gene Therapy Institute, Oregon Health & Science University, Beaverton, OR, USA. [12]Center for Global Infectious Disease Research, Seattle Children's Research Institute, Seattle, WA, USA. [13]Department of Pediatrics, University of Washington, Seattle, WA, USA. [14]InterRayBio LLC, Cleveland, USA. [15]Veterans Affairs Medical Center, Cleveland, OH, USA. [16]These authors contributed equally: Anna C. Winnicki, Melanie H. Dietrich. [17]These authors jointly supervised this work: Jürgen Bosch, Christopher L. King. ✉e-mail: jbosch@case.edu; cxk21@case.edu

