## [Peer Review file · Nature Communications]

Potent AMA1-specific human monoclonal antibody against *P. vivax* Pre-erythrocytic and Blood Stages

Corresponding Author: Professor Christopher King

Version 0:

Reviewer comments:

Reviewer #1

(Remarks to the Author)

Summary:

This manuscript reports the isolation of a monoclonal antibody that inhibits binding of *Plasmodium vivax* (Pv) Apical Membrane Antigen 1 (AMA-1). Neutralization of this antigen has been reported to neutralize both blood stages as well as stopping sporozoite invasion of hepatocytes, thereby, neutralizing the develop of liver-stages. The latter is essential to prevent the formation of hypnozoites, which are the prime drivers of infections and transmission of Pv in endemic areas. Thus, there is significant interest in identifying monoclonal antibodies that could be used to reduce the hypnozoite reservoir and reduce disease and transmission. This antibody, in principle, could reduce hypnozoite burden and may have the added benefit of neutralizing blood-stage parasites if used in vivo. Together, these make the work presented here important to the field of Pv research.

In the manuscript, the authors detail the isolation of the antibody through standard monoclonal methods, assessment of neutralization of blood and liver-stages, and also perform structural analysis to propose mechanisms that result in the best antibody's ability to neutralize. The latter is particular useful for the field as PvAMA-1 has been a vaccine candidate, but due to its polymorphic nature has not provided significant protection from blood-stage infection. Thus, the structural analyses provide some insights into how antibodies targeting these regions can be evaded, which may lead to improvement of AMA-1 immunogen design. While this work is important, there are a number of concerns that reduce the impact of the manuscript.

The major weaknesses are as follows:

1. The authors provide no data on how well this antibody will perform in vivo, which the authors indicate could be tested using humanized mice with Pv or potentially *P. cynomolgi*. Given the IC50 values reported in the paper, this would be needed to truly assess the impact and therapeutic potential of this antibody.
2. Representative data demonstrating the isolation of the Pv AMA-1 + memory B cells is not presented for assessment in the manuscript. This is standard in most manuscripts assessing or creating monoclonal antibodies. This is more important here because there is some concern that the antibodies used in the panel may have interfered with the binding of the B cell receptor to the antigen due to steric hinderance, thereby, reducing the number of clones identified.
3. The clonal analysis is not consistent with field standards. Generally, to be considered a clone, a B cell must have the same VDJ segment usage, have the same CDR3 length, and 85% AA identity across heavy and light chains. The rationale for using the term clonal group is unclear in the manuscript, and there are also standards for tracing lineages of B cells. Additionally, it seems that not many high-quality sequences were obtained. An example of a suite of software the authors might find useful for better assessing this information is Immcantation, which can be found here <https://immcantation.readthedocs.io/en/stable/index.html>
4. There are significant discrepancies between the authors use of terms, language, and numbers in the main text, figure legends, and methods. For example, the authors indicate in the abstract that they isolated antibodies from the plasma, but their methods describe isolation from peripheral blood mononuclear cells. Additionally, the text says that *Anopheles dirus* were fed directly on patients to generate Pv sporozoites in Thailand while the methods indicate the mosquitoes were fed blood from infected patients. These inconsistencies should be reviewed and fixed.

Minor Weaknesses

1. The authors indicate in their results that they 'may have a strain transcending antibody', but then, they state it is strain-transcending in the discussion. One phrase should be selected. Also, strains are typically associated with the laboratory and continuous maintenance of a parasite versus what was assessed here is Pv isolates/variants since these are circulating parasites that cannot be obtained again.
2. It should be made clear the parasitemia and liver-stage form cut-offs used to assess if the data from the neutralization experiments were valid. Currently, there is information suggesting there were assessments, but it is important to state what the experiment had to be for you to include the data in the analysis (i.e. what were the lower limits of quantitation).
3. In the legend for Figure 1 and elsewhere, the authors should consider using the field-standard abbreviations/capitalizations for antibody chains and Kd. Additionally, it would be helpful to indicate whether the number of SHM indicated in Fig. 1A are amino acid or nucleotide mutations.
4. In the legend for Figure 3B, the lowest IC50 measured is listed as 0.07 µg/mL. This does not match the text, which lists the lowest IC50 measured as 0.01 µg/mL.
5. In Figure 4B, it would be helpful and improve ease of understanding to shade the buried surface area on the structure model. Similarly, changing the legend labels from the PDB IDs to the protein names used elsewhere in the text would improve clarity for the reader.
6. Generally speaking, it would improve clarity if the numerical orientation of the x-axis were kept constant between figures.

Reviewer #2

(Remarks to the Author)

Winnicki and colleagues report on the isolation and characterization of human monoclonal antibodies (mAbs) against *Plasmodium vivax* (Pv) AMA1. AMA1 has a somewhat spotty record historically in vaccine development for *P. falciparum* (Pf), thought to be inhibited by the extensive polymorphisms among circulating strains. Here the authors evaluate mAbs to Pv, which historically is less studied for vaccine development. After screening serum from infected subjects, they isolated B cells reactive to PvAMA1. They evaluated their ability to bind to AMA1 from other strains, observing extensive cross-reactivity to another Pv strain and to PkAMA1, and some limited cross-reactivity to PfAMA1. The leading antibody, 826827, showed inhibitory activity against both blood and pre-erythrocytic stages, across multiple clinical isolates. Potency against field isolates for hepatocyte invasion was similar to that of the prototype anti-CSP mAb 2F2. Structural studies revealed that the antibody binds to the protein element that interacts with RON2, a part of AMA1 that is less polymorphic. Thus, mechanistically, the antibody likely inhibits invasion of Hepatocytes or RBCs by occupying the hydrophobic groove that would normally be occupied by RON2 during invasion.

As a whole, this is a well-presented study, with data that strongly support their overall conclusions. Strain transcendence is an important aspect, which is highly relevant to vaccine design. The blocking of both PE and BS invasion is highly interesting, and the potency with which it blocks is extraordinary (in fact, perhaps under sold in the manuscript). In fact, the potential broader impact of this finding is also a bit undersold and could be hyped a bit more. This study also touches upon the importance of affinity in mediating potent blocking activity, a feature that is becoming more and more clear to be critical for functional blocking. Overall, despite the conventional wisdom that AMA1 is not a great vaccine target for Pf, this study highlights that there are elements within the protein that could be highly effective vaccine targets. As such, this study may guide structure-based design toward a vaccine candidate focused on the RON2 binding groove, perhaps in combination with other potent targets, that ultimately may be able to reduce or eliminate transmission of *P. vivax*.

Some minor comments:

- Fig 2A seems unnecessary. It is in a highly contrived model, and IC50 values were more potent in the biologically relevant reticulocyte invasion assay anyway.
- There is some mixing of equivalent units, ng/ul and ug/ml. I'd pick one and unify for readability.
- Some grammatical errors need to be fixed throughout.
- A little more experimental detail on binding would be useful in the results. The assay used in binding studies does not appear to be noted in the text (results section 1, paragraph 2).
- Similar to previous, a touch more info on the assays in the body of the text would go a long way for broad readership.
- Other than antigen binding, is the epitope specificity known for any other clones? It would be interesting to understand the antibody within the context of the rest of the anti-AMA1 response. E.g., is it a rare target or immunodominant, etc.

Reviewer #3

(Remarks to the Author)

In this manuscript, Winnicki et al. isolate human mAbs that target the Plasmodium vivax protein AMA1, a protein important for invasion of the parasite into hepatocytes and reticulocytes. They show that one antibody, 826827, performed better than others at blocking erythrocytes using a chimeric P. falciparum line expressing PvAMA1. 826827 was also able to inhibit hepatocyte invasions at lower IC50 than all others tested in vitro. A crystal structure of the 826927 Fab was solved bound to PvAMA1 and showed that the heavy chain of 826827 has an extended CDRH3 that binds in a hydrophobic groove that serves as the binding site of RON2, providing a mechanism for 826927's inhibition. The interaction of AMA1 and RON2 is important for both sporozoite and merozoite invasion. They further show that the key interactions in the 826827-PvAMA1 have low polymorphisms among clinical isolates. In all, this study provides important biological and structural information for P. vivax immunology and could be useful for vaccine design.

Major comments:

In the intro the authors mention that "we have identified a potent, strain-transcending humAb that blocks RON2-loop binding and can inhibit both blood stage and sporozoite infection." I believe this manuscript would benefit from an in vivo experiment for liver burden, either using the human liver chimeric mouse model or with a chimeric P. berghei expressing PvCSP since they mention the humAb also inhibit sporozoite infection. Have the authors tested 826827 binding to PvAMA1? The authors specifically looked for "PvRON2-binding-inhibition" huMabs but it is possible that there are huMabs that will prevent AMA-1 action by not blocking RON2. Have the authors looked at this at all? I think it should be added to the discussion. This recent paper (<https://www.ncbi.nlm.nih.gov/pmc/articles/PMC10475129/>) showed that antibodies that do not inhibit AMA1-RON2 interaction are highly protective.

Minor comments:

In the introduction, the term humAbs is used multiple times before it is defined in the last paragraph and on one occasion "human humAbs" is used. However, it is clearly defined in the abstract. Maybe revise the intro.

It is not clear from the beginning what the naming convention of the humAbs is. I was initially confused when the heavy chain was referred to as 826. This should be clarified. It seems indeed that 826827 comes from 826HC and 827LC but this is not mentioned in the results. Usually one refers to the Mab 826827 HC and LC otherwise it seems they come from a different antibody.

The buried surface area of 826827 is stated twice (page 8 and page 10 in the PDF) in the text and with different significant figures. In general, buried surface area are approximate – at place the authors mention 1392.5 Å² and other 1392 Å². Maybe just change to "˜1392 Å²."

Rewrite this sentence: "The combined buried surface area is 1392.5 Å² large, while the major contribution of 70% is provided by the heavy chain 826." To something like "The major contribution of humAb 826827 is through its heavy chain (˜70% of the ˜1392 Å² buried surface area)." Also that sentence can be moved when describing the interactions first. It seems a bit out of place to have a repeat of this information in another paragraph.

Suggestion: "Figure 5A shows that the contact residues between PvAMA1 and RON2-loop overlap with the residues of PvAMA1 that contact humAb 826827 CDR-H3 residues, which contributes to ˜X % of the total buried surface area of 826827 interaction with PvAMA1". (it is actually unclear what is the CDRH3 only contribution to the interaction the way it is currently written).

Same here "Comparing the RON2 and 826 CDR3 binding site to other available Plasmodium species and model systems reveals that P..." Since CDRL3 is not involved at all in the interaction specify CDRH3.

Rewrite sentence: "They are also exhibit similar potency compared to a PvAMA1 humAb that was previously produced based on an IgG"

Figure 1 – is the number of SHM amino acid or nucleotide based?

Figure 2A, 2C, and 3B, error bars should be added to show distribution of the replicates.

Reviewer #4

(Remarks to the Author)

Reviewer #5

(Remarks to the Author)

Version 1:

Reviewer comments:

Reviewer #1

(Remarks to the Author)

The revised manuscript is much improved, and the results are interesting. Importantly, I think that the comparison of the different assays helps to start defining an appropriate pipeline for testing monoclonal antibodies for Pv.

Fig 3A and 3B – Why are the data shown differently in these panels? They are generally the same assay, albeit with different hepatocytes. However, IC50 should be able to be calculated for either or inhibition to show consistency. It would also help compare the assays head-to-head for determining appropriate pipelines for monoclonal antibodies in the future.

Fig 3 – The liver stages still leaves a few behind based on supplementary data in Figure s6. Can you comment directly in discussion how this may impact the utility or if alternative approaches could be used?

Line 194 – says 'reticulates' instead of reticulocytes

Reviewer #2

(Remarks to the Author)

This reviewer had few major criticisms, all of which have been responded to, if not fully addressed. Other reviewer comments have been addressed, and the authors added a major new experiment in FRG mice to assess blocking activity in vivo against field isolates. This experiment shows blocking activity of the mAb in vivo against the sporozoite, and represents a major investment for this study, since infected mosquitos are shipped from Thailand and each mouse is nearly the cost of a non-human primate. The data support the overall conclusions and have strengthened the manuscript.

Reviewer #3

(Remarks to the Author)

The authors have addressed the reviewers' comments.

Reviewer #4

(Remarks to the Author)

Below is a point-by-point response to the reviewers:

Reviewer #1

1. The authors provide no data on how well this antibody will perform in vivo, which the authors indicate could be tested using humanized mice with Pv or potentially P. cynomolgi. Given the IC50 values reported in the paper, this would be needed to truly assess the impact and therapeutic potential of this antibody.

*Response: We agree that data from animal models may help further understand the efficacy and potential prophylaxis and therapeutic potential of the monoclonal antibodies. The availability of animal models for P. vivax is extremely limited. Evaluating the MAbs in primate models will need to be a long-term goal and is not possible within the scope and timeframe of this paper; access to primate models is limited and their use is ethically complex and costly. However, we have been able to conduct experiments using human liver chimeric mice (FRGhuHep), which have been transplanted with human hepatocytes and support P. vivax sporozoite infection and liver-stage development, (Mikolajczak et al., 2015). Using this model we found that administration of humAb 826827 prior to infection challenge markedly reduced the liver burden of P. vivax infection, as shown in our revised Figure 4 and Supplemental Figure S5. This is consistent with our in vitro data showing that humAb 826827 inhibited infection of hepatocytes by P. vivax sporozoites. Results are shown **Figure 4 and Supplemental Figure 6**. New sections describing these results are detailed in **lines 210-227**.*

2. Representative data demonstrating the isolation of the Pv AMA-1 + memory B cells is not presented for assessment in the manuscript. This is standard in most manuscripts assessing or creating monoclonal antibodies. This is more important here because there is some concern that the antibodies used in the panel may have interfered with the binding of the B cell receptor to the antigen due to steric hinderance, thereby, reducing the number of clones identified.

*Response: We show flow panels from which the PvAMA1-specific B cells were isolated in **supplemental Figure S2**. The PvAMA1-specific B cells were tightly clustered and well separated from the non-antigen-specific B cells, suggesting strong binding of the PvAMA1-specific tetramers.*

3. The clonal analysis is not consistent with field standards. Generally, to be considered a clone, a B cell must have the same VDJ segment usage, have the same CDR3 length, and 85% AA identity across heavy and light chains. The rationale for using the term clonal group is unclear in the manuscript, and there are also standards for tracing lineages of B cells. Additionally, it seems that not many high-quality sequences were obtained.

*Response: We appreciate this comment from the reviewer and as suggested we have redefined the clonal groups to have the same VDJ segment and CDR3 length and 85% or greater amino acid similarity for CDR3. With this modified criterion, we now have 67 clonal groups, and the number of B cells identified in each clonal group has changed for some clonal groups, as **shown in Figure 1A, Supplemental Figure 2 and detailed in lines 141-146**. We provide the VDJ usage and CDR3 sequences for all 158 isolated B cells in **Supplemental Table 1**. In terms of the lower number of high-quality sequences, due to the primer set used for single-cell sorting of antigen-specific B cells, obtaining high-quality sequences across the entire*

heavy and light chains is at times challenging. Nonetheless, CDR3 sequences consistently maintain high quality.

4. There are significant discrepancies between the authors use of terms, language, and numbers in the main text, figure legends, and methods. For example, the authors indicate in the abstract that they isolated antibodies from the plasma, but their methods describe isolation from peripheral blood mononuclear cells. Additionally, the text says that *Anopheles dirus* were fed directly on patients to generate Pv sporozoites in Thailand while the methods indicate the mosquitoes were fed blood from infected patients. These inconsistencies should be reviewed and fixed.

Response: Thank you for pointing out these inconsistencies. The abstract has been changed to PBMC, and the text has been changed to designate that mosquitoes were fed on human blood, not directly on humans. This is consistent with the methods sections. Some other discrepancies have been identified and corrected.

Minor Weaknesses

1. The authors indicate in their results that they 'may have a strain transcending antibody', but then, they state it is strain-transcending in the discussion. One phrase should be selected. Also, strains are typically associated with the laboratory and continuous maintenance of a parasite versus what was assessed here is Pv isolates/variants since these are circulating parasites that cannot be obtained again.

Response: We now say recognizing and active against multiple clinical isolates, rather than strain-transcending.

2. It should be made clear the parasitemia and liver-stage form cut-offs used to assess if the data from the neutralization experiments were valid. Currently, there is information suggesting there were assessments, but it is important to state what the experiment had to be for you to include the data in the analysis (i.e. what were the lower limits of quantitation).

*Response: The threshold for *P. vivax* invasion of reticulocytes in controls is 0.2% parasitemia (**detailed in lines 666-667**). This clearly separates new invasion events from uninfected reticulocytes on FACS analysis. For the experiments performed with AMA1 humAbs, invasion rates in controls varied from 0.24 to 0.7% (**detailed in lines 733-735**). For the liver stage assay, 60 liver stage parasites/well in controls are used as cut-offs for a valid experiment. The mean of invasion controls varied from 72 to 904 for experiments using humAbs to PvAMA1.*

3. In the legend for Figure 1 and elsewhere, the authors should consider using the field-standard abbreviations/capitalizations for antibody chains and Kd. Additionally, it would be helpful to indicate whether the number of SHM indicated in Fig. 1A are amino acid or nucleotide mutations.

*Response: As shown in **figure 1A**, we have used the standard nomenclature in the legend for antibody chains. We now designate SHM as nucleotide mutations, **detailed in lines 1118-1119**.*

4. In the legend for Figure 3B, the lowest IC50 measured is listed as 0.07 µg/mL. This does not match the text, which lists the lowest IC50 measured as 0.01 µg/mL.

Response: In **Figure 3A**, one isolate's IC50 is measured as 0.01 $\mu\text{g}/\text{mL}$. However, **3B** shows 3 additional isolates with detailed IC50 curves, and the lowest is 0.07 $\mu\text{g}/\text{mL}$. The text (**lines 1160-1163**) has been corrected to reflect the lowest IC50 in 3B as 0.07 $\mu\text{g}/\text{mL}$.

5. In Figure 4B, it would be helpful and improve ease of understanding to shade the buried surface area on the structure model. Similarly, changing the legend labels from the PDB IDs to the protein names used elsewhere in the text would improve clarity for the reader.

Response: We appreciate the suggestion of the reviewer but found that shading the epitope made **panel B** busier and more confusing. We also tried to highlight the epitope with an outline instead (please, see comparisons below) but still think that the original panel B shows best what we want to show. We changed the legends in the Figure and included the protein names as suggested by the reviewer. In addition we included a new **Supplementary Figure 10** where the epitope and paratopes are shaded on the protein surfaces.

Original panel B

Panel B with shading

Panel B with outline

A PvAMA1

B 826827

6. Generally speaking, keeping the numerical orientation of the x-axis constant between figures would improve clarity.

Response: All x-axes on the graphs are oriented similarly, with the largest concentration starting on the left side and decreasing in concentration as it moves to the right. Some graphs start at 1000 $\mu\text{g}/\text{mL}$ of humAb, whereas others start at 100 $\mu\text{g}/\text{mL}$, reflecting different experimental conditions with different assays.

Reviewer #2

Overall Response: Thank you for your overall positive response to the manuscript and constructive comments.

Some minor comments:

- Fig 2A seems unnecessary. It is in a highly contrived model, and IC50 values were more potent in the biologically relevant reticulocyte invasion assay anyway.

*Response: We prefer to retain **Figure 2A** as valuable data and complements data generated using *P. vivax* isolates; the other reviewers have not requested removal of that figure. The advantage of the Pf-PvAMA1 transgenic parasite model is that it expresses the Palo Alto variant of PvAMA1, which is the same PvAMA1 allele that was used to isolate the humAbs. Therefore, we could evaluate the activity of the humAbs without the potential confounding effect of polymorphisms in PvAMA1 that occur in clinical isolates.*

- There is some mixing of equivalent units, ng/ul and ug/ml. I'd pick one and unify for readability.

*Response: ng/ml is used only in **Figure 1** because reactivity to recombinant protein is much more sensitive compared to the amount of antibody required for invasion assays. We changed this to $\mu\text{g}/\text{mL}$ in **Figure 1** legend.*

- Some grammatical errors need to be fixed throughout.

Response: Thank you. We edited the text more carefully.

- A little more experimental detail on binding would be useful in the results. The assay used in binding studies does not appear to be noted in the text (results section 1, paragraph 2).

- Similar to previous, a touch more info on the assays in the body of the text would go a long way for broad readership.

Response: We have further described key assay methods in the result sections

- Other than antigen binding, is the epitope specificity known for any other clones? It would be interesting to understand the antibody within the context of the rest of the anti-AMA1 response. E.g., is it a rare target or immunodominant, etc.

Response: This is an interesting point. A future goal will be to co-crystallize other human monoclonal antibodies. We will use the RON2—PvAMA1 binding assay to identify Pv-exposed individuals with blocking activity, like the individual from whom we isolated the most inhibitory monoclonal antibody in the present study. We will then use our different monoclonals to see if they compete with the inhibitory activity in sera. This will determine whether this response is rare or immunodominant. These experiments will be part of a future project, which we plan to publish as a separate manuscript.

Reviewer # 3

Major comments:

In the intro the authors mention that “we have identified a potent, strain-transcending humAb that blocks RON2-loop binding and can inhibit both blood stage and sporozoite infection.” I believe this manuscript would benefit from an in vivo experiment for liver burden, either using the human liver chimeric mouse model or with a chimeric *P. berghei* expressing PvCSP since they mention the humAb also inhibit sporozoite infection. Have the authors tested 826827 binding to PbAMA1?

*Response: We have shown humAb 826827 inhibits sporozoite invasion in the human liver chimeric mouse model, which is now included in the manuscript. However, based on structural and sequence analysis, we suspect that 826827 would not show as potent inhibition of P. berghei parasites. There is 45% identity for amino acids in the PvAMA1 epitope recognized by 826827 between Pb and Pv. The amino acid identity for Pv and PfAMA1 (31.5%), and 826827 does not bind to PfAMA1. Thus, 826827 is unlikely to recognize PbAMA1, although we have not tested it. Results for in vivo data are shown **Figure 4 and Supplemental Figure 6**. New sections describing these results are detailed in **lines 210-227**.*

The authors specifically looked for “PvRON2-binding-inhibition” huMabs but it is possible that there are huMabs that will prevent AMA-1 action by not blocking RON2. Have the authors looked at this at all? I think it should be added to the discussion. This recent paper (<https://www.ncbi.nlm.nih.gov/pmc/articles/PMC10475129/>) showed that antibodies that do not inhibit AMA1-RON2 interaction are highly protective.

Responses: This is an interesting point, and we have considered it. Most of the humAbs generated from this donor appear to have some blocking activity in PvAMA1-RON2 assay. No humAbs antibodies that do not block or weakly block PvAMA1-RON2 demonstrate potent invasion inhibition in vitro using clinical isolates. We may have biased selection of PvAMA1-RON2 blocking Abs because this individual had potent blocking activity in serum. To address this question, we are isolating humAbs to PvAMA1 from other individuals, some without detectable PvAMA1-RON2 blocking activity. We address this point briefly in the discussion.

Minor comments:

In the introduction, the term humAbs is used multiple times before it is defined in the last paragraph and on one occasion “human humAbs” is used. However, it is clearly defined in the abstract. Maybe revise the intro.

Response: Corrected

It is not clear from the beginning what the naming convention of the humAbs is. I was initially confused when the heavy chain was referred to as 826. This should be clarified. It seems indeed that 826827 comes from 826HC and 827LC but this is not mentioned in the results. Usually one refers to the Mab 826827 HC and LC otherwise it seems they come from a different antibody.

*Response: This humAb numbering is now described at the binding of the results section (**lines 151-152**).*

The buried surface area of 826827 is stated twice (page 8 and page 10 in the PDF) in the text and with different significant figures. In general, buried surface area are approximate – at place the authors mention 1392.5 Å² and other 1392 Å². Maybe just change to “~1392 Å².”

Rewrite this sentence: “The combined buried surface area is 1392.5 Å² large, while the major contribution of 70% is provided by the heavy chain 826.” To something like “The major contribution of humAb 826827 is through its heavy chain (~70% of the ~1392 Å² buried surface area).” Also that sentence can be moved when describing the interactions first. It seems a bit out of place to have a repeat of this information in another paragraph.

Suggestion: “Figure 5A shows that the contact residues between PvAMA1 and RON2-loop overlap with the residues of PvAMA1 that contact humAb 826827 CDR-H3 residues, which contributes to ~X % of the total buried surface area of 826827 interaction with PvAMA1”. (it is actually unclear what is the CDRH3 only contribution to the interaction the way it is currently written).

Same here “Comparing the RON2 and 826 CDR3 binding site to other available Plasmodium species and model systems reveals that P...” Since CDRL3 is not involved at all in the interaction specify CDRH3.

Rewrite sentence: “They also exhibit similar potency compared to a PfAMA1 humAb that was previously produced based on an IgG”

Response: We agree with the reviewer and now only mention the buried surface area once as an approximate and have moved the information of the CDR-H3 contribution to the section when first describing the interaction.

In line 236-238, we changed the sentence “Five of the six complementarity-determining regions (CDR, namely: L1, L2, H1, H2, and H3) form direct contacts with PvAMA1 with a buried interaction surface of 1392 Å², with the light chain contributing 432 Å² and the heavy chain contributing 959 Å²” to “Five of the six complementarity-determining regions (CDR, namely: L1, L2, H1, H2, and H3) form direct contacts with PvAMA1 with a buried interaction surface of ~1392 Å², with the CDR-H3 loop of 826827 contributing 70% of the buried surface area (**Figure 5B**)..”

As suggested by the reviewer we changed the sentence in **line 281-285** to specify CDR-H3. It reads now: “Comparing the RON2 and CDR-H3 binding site of 826827 to other available Plasmodium species and model systems reveals that *P. cynomolgi* AMA1 is 100% conserved and would therefore serve as a predictive non-human primate model for Pv challenge infections to evaluate humAb 826827 (**Figure 6A, Supplemental Figure 8**).

Line 341-343: We rewrote the sentence “They also exhibit similar potency compared to a PfAMA1 humAb that was previously produced based on an IgG...”, to “They exhibit similar potency to a PfAMA1 humAb produced from an IgG sequence isolated from a Ghanaian with an IC₅₀ of 35 µg/mL against the Pf 3D7 variant in vitro.”

Figure 1 – is the number of SHM amino acid or nucleotide based?

Response: Addressed in figure legend (lines 1118-1119).

Figure 2A, 2C, and 3B, error bars should be added to show distribution of the replicates.

*Response: Corrected, however the graph in **Figure 2C** is displaying the dose curve for each replicate that was performed. Therefore, this graph represents the distribution of replicates without the need for error bars.*

REVIEWERS' COMMENTS

Reviewer #1 (Remarks to the Author):

The revised manuscript is much improved, and the results are interesting. Importantly, I think that the comparison of the different assays helps to start defining an appropriate pipeline for testing monoclonal antibodies for Pv.

Fig 3A and 3B – Why are the data shown differently in these panels? They are generally the same assay, albeit with different hepatocytes. However, IC50 should be able to be calculated for either or inhibition to show consistency. It would also help compare the assays head-to-head for determining appropriate pipelines for monoclonal antibodies in the future.

Authors Response:

We have removed Figure 3B as suggested by the reviewer.

Fig 3 – The liver stages still leaves a few behind based on supplementary data in Figure s6. Can you comment directly in discussion how this may impact the utility or if alternative approaches could be used?

Authors Response:

We have added a few lines of clarifications and explanation in the text to help explain why there is signal in the qPCR results for the *in vivo* experiments as well as added one more reference demonstrating the same finding with Pf in this model system.

Line 194 – says ‘reticulates’ instead of reticulocytes

Authors Response:

This spelling error has been fixed and we thank the reviewer for catching our error.

Reviewer #2 (Remarks to the Author):

This reviewer had few major criticisms, all of which have been responded to, if not fully addressed. Other reviewer comments have been addressed, and the authors added a major new experiment in FRG mice to assess blocking activity *in vivo* against field isolates. This experiment shows blocking activity of the mAb *in vivo* against the sporozoite, and represents a major investment for this study, since infected mosquitos are shipped from Thailand and each mouse is nearly the cost of a non-human primate. The data support the overall conclusions and have strengthened the manuscript.

Authors Response:

We thank the reviewer for helping us improve the manuscript.

Reviewer #3 (Remarks to the Author):

The authors have addressed the reviewers' comments.

Authors Response:

We thank the reviewer for helping us improve the manuscript.

Reviewer #4 (Remarks to the Author):

Authors Response:

We thank the reviewer for helping us improve the manuscript.